# Transsynaptic mapping of *Drosophila* mushroom body output neurons

Kristin M Scaplen[1,2,3]*, Mustafa Talay[1†], John D Fisher[1], Raphael Cohn[4‡], Altar Sorkaç[1], Yoshi Aso[5], Gilad Barnea[1]*, Karla R Kaun[1]*

[1]Department of Neuroscience, Brown University, Providence, United States; [2]Department of Psychology, Bryant University, Smithfield, United States; [3]Center for Health and Behavioral Sciences, Bryant University, Smithfield, United States; [4]Laboratory of Neurophysiology and Behavior, The Rockefeller University, New York, United States; [5]Janelia Research Campus, Howard Hughes Medical Institute, Ashburn, United States

**\*For correspondence:**
kscaplen@bryant.edu (KMS);
gilad_barnea@brown.edu (GB);
karla_kaun@brown.edu (KRK)

**Present address:** [†]Howard Hughes Medical Institute, Department of Molecular and Cellular Biology, Harvard University, Cambridge, United States; [‡]Department of Biological Studies, Columbia University, New York, United States

**Competing interests:** The authors declare that no competing interests exist.

**Abstract** The mushroom body (MB) is a well-characterized associative memory structure within the *Drosophila* brain. Analyzing MB connectivity using multiple approaches is critical for understanding the functional implications of this structure. Using the genetic anterograde transsynaptic tracing tool, *trans*-Tango, we identified divergent projections across the brain and convergent downstream targets of the MB output neurons (MBONs). Our analysis revealed at least three separate targets that receive convergent input from MBONs: other MBONs, the fan-shaped body (FSB), and the lateral accessory lobe (LAL). We describe, both anatomically and functionally, a multilayer circuit in which inhibitory and excitatory MBONs converge on the same genetic subset of FSB and LAL neurons. This circuit architecture enables the brain to update and integrate information with previous experience before executing appropriate behavioral responses. Our use of *trans*-Tango provides a genetically accessible anatomical framework for investigating the functional relevance of components within these complex and interconnected circuits.

## Introduction

Neural circuits underlie all brain functions, from sensation and perception to learning, memory, and behavior. One of the greatest scientific challenges is to understand how neural circuits are structurally and functionally connected to support the extensive repertoire of behaviors animals use to interact with the world. *Drosophila melanogaster* is a powerful model for mapping the fundamental architecture of neural circuit organization in the context of specific behaviors due to its complex yet tractable brain. With a nervous system of approximately 100,000 neurons and a rich genetic toolkit that offers the potential to selectively manipulate subsets of neurons in behaving animals, significant effort has been devoted to establishing a detailed map of structural neural connectivity in the fly in an effort to then layer on function (*Aso et al., 2014a*; *Bates et al., 2020*; *Couto et al., 2005*; *Deng et al., 2019*; *Eichler et al., 2017*; *Eschbach et al., 2020*; *Fishilevich and Vosshall, 2005*; *Frechter et al., 2019*; *Grabe et al., 2015*; *Kondo et al., 2020*; *Li et al., 2020*; *Marin et al., 2020*; *Otto et al., 2020*; *Peng et al., 2011*; *Shao et al., 2014*; *Takemura et al., 2017*; *Zheng et al., 2018*). However, establishing a map of connectivity has proven to be a monumental task. Here, we bypass time and manpower by mapping mushroom body (MB) neural circuits across multiple animals using the recently developed genetic anterograde tracing tool *trans*-Tango (*Talay et al., 2017*). In *trans*-Tango, a synthetic signaling pathway converts the activation of a cell surface receptor into expression of a reporter gene via site-specific proteolysis. This pathway is introduced into all neurons while the starter neurons of interest express the ligand that activates the pathway and present it in their synapses. Binding of the ligand to its receptor on the postsynaptic partners activates the

signaling pathway and leads to expression of a reporter that selectively labels these postsynaptic neurons (*Talay et al., 2017*).

The insect MB is a prominent neuropil structure that integrates inputs from multiple sensory modalities (*Caron et al., 2013*; *Ehmer and Gronenberg, 2002*; *Gruntman and Turner, 2013*; *Li and Strausfeld, 1997*; *Li and Strausfeld, 1999*; *Liu et al., 2006*; *Liu et al., 2016*; *Marin et al., 2020*; *Marin et al., 2002*; *Schildberger, 1984*; *Strausfeld and Li, 1999a*; *Strausfeld and Li, 1999b*; *Vogt et al., 2016*; *Vogt et al., 2014*; *Wang et al., 2016*; *Yagi et al., 2016*; *Zars, 2000*) and has a well-established role in learning and memory (*Davis, 1993*; *de Belle and Heisenberg, 1994*; *Heisenberg, 1998*; *Heisenberg, 2003*; *Heisenberg et al., 1985*; *Pascual and Préat, 2001*; *Zars et al., 2000*). The MB comprises thousands of densely packed Kenyon cell neural fibers that are organized into three separate lobes (α/β, α′/β′, and γ). Kenyon cell neural fibers form *en passant* synapses along the length of their axons with efferent cells called MB output neurons (MBONs; *Aso et al., 2014a*; *Eichler et al., 2017*; *Eschbach et al., 2020*; *Li and Strausfeld, 1997*; *Li and Strausfeld, 1999*; *Mobbs, 1982*; *Takemura et al., 2017*). In addition to receiving processed sensory information, the MB integrates valence-related input from dopamine neurons (DANs; *Aso et al., 2014a*; *Eichler et al., 2017*; *Eschbach et al., 2020*; *Takemura et al., 2017*). This architecture positions the MB as a high-level integration center for the representations of multisensory cues and their perceived valence. Thus, the MB is an ideal neural structure for mapping structural connectivity and inferring fundamental architecture of neural circuits in the context of defined inputs and outputs across species.

Early neuroanatomical and functional work in insects described distinct organization within the MB's afferent and efferent innervation patterns (*Ito et al., 1998*; *Li and Strausfeld, 1997*; *Li and Strausfeld, 1999*; *Mao and Davis, 2009*; *Nässel and Elekes, 1992*; *Tanaka et al., 2008*; *Waddell, 2013*). A more refined analysis of the neural circuits associated with the *Drosophila* MB was recently achieved through the use of split-Gal4 lines that enabled selective genetic access to specific neuronal populations (*Aso et al., 2014a*). These delineate a compartmentalization of the MB lobes by overlapping patterns of innervating DANs and MBONs (*Aso et al., 2014a*; *Eichler et al., 2017*; *Eschbach et al., 2020*; *Takemura et al., 2017*). Projections from the MBONs terminate within discrete neuropils, including the lateral horn (LH), crepine (CRE), superior medial (SMP), intermediate (SIP), and lateral (SLP) protocerebrum (*Aso et al., 2014a*; *Ito et al., 2014*). These neuropils have also been described as convergence sites of MBONs as different MBONs send converging outputs to similar subregions in these structures (*Aso et al., 2014a*; *Ito et al., 2014*). Within these neuropils, evidence suggests that MBON axons synapse onto dendrites of DANs and other MBONs providing opportunities for feedback to the MB (*Aso et al., 2014a*; *Eichler et al., 2017*; *Scaplen et al., 2020*). Evidence also suggests MBON axons synapse onto dendrites of neurons projecting to other structures, including the FSB (*Aso et al., 2014a*; *Eichler et al., 2017*; *Scaplen et al., 2020*). Additionally, similar to other insects and to the first instar *Drosophila* larva, MBONs in the adult brain are hypothesized to synapse on local interneurons whose processes are confined to the limits of the target neuropil but play a role in modulating input and output signals (*Eichler et al., 2017*; *Phillips-Portillo and Strausfeld, 2012*). These convergent neuropils, however, are characterized by highly complex arborizations of dendrites and axons. Therefore, identifying the specific neural components that receive synaptic input from various MBONs is challenging.

Postsynaptic partners of specific neurons were initially identified by mapping the movement of cobalt ions from one neuron into another (*Strausfeld and Obermayer, 1976*). Later, candidate synaptic partners were identified either through the use of computational approaches to reveal overlapping arborization patterns or using molecular techniques such as fluorescent protein reconstitution across neurons (*Chiang et al., 2011*; *Feinberg et al., 2008*; *Jefferis et al., 2007*; *Li et al., 2016*; *Lin et al., 2013*; *Macpherson et al., 2015*; *Shearin et al., 2018*; *Wolff et al., 2015*). Recently, much effort has been devoted to map synaptic connections across the fly brain using whole brain serial electron microscopy (EM; *Li et al., 2020*; *Ohyama et al., 2015*; *Schneider-Mizell et al., 2016*; *Xu et al., 2020*; *Zheng et al., 2017*; *Zheng et al., 2018*). Although EM reconstruction offers synaptic structural resolution, it is labor intensive and it does not account for the synaptic strength nor the potential variability in synaptic connectivity across animals. We sought to test previous predictions regarding MBON connectivity (*Aso et al., 2014a*) and complement the EM anatomic data by mapping the postsynaptic partners of all MBONs using the genetic anterograde transsynaptic tracing tool, *trans*-Tango (*Talay et al., 2017*). We found that MBONs have a broad reach in their spread of

postsynaptic connections. We observed abundant interconnectivity as previously predicted, with MBONs synapsing on DANs, and several MBONs converging on other MBONs. Further, we confirmed direct connections between the MBONs and two additional regions, the fan-shaped body (FSB) and the lateral accessory lobe (LAL). We identified, both anatomically and functionally, a multilayer circuit that includes GABAergic and cholinergic MBONs that converge on the same subset of FSB and LAL postsynaptic neurons. This circuit architecture provides an opportunity to integrate information processing before executing behavior, and we propose that multilevel integration across brain regions is critical for updating information processing and memory.

## Results

### Divergence and convergence of the MBONs circuits

Circuit convergence, divergence, and re-convergence can be found throughout the nervous systems of both invertebrates and vertebrates and play a pivotal role in providing behavioral flexibility (*Eschbach et al., 2020*; *Jeanne and Wilson, 2015*; *Man et al., 2013*; *Miroschnikow et al., 2018*; *Mišić et al., 2014*; *Ohyama et al., 2015*). Given the importance of the MBONs in driving behavioral choice, we first sought to reveal patterns of divergence and convergence by identifying the postsynaptic connections of the MBONs innervating each of the 15 MB compartments using *trans*-Tango (*Talay et al., 2017*). Since *trans*-Tango signal depends on the strength and specificity of the GAL4 driver being used, we selected 28 previously published MBON split-GAL4 lines specific to individual MBONs, or sparse but overlapping subsets of MBONs (*Aso et al., 2014a*). We combined *trans*-Tango with chemogenetic active zone marker using the *brp-SNAP* knock-in to increase uniformity of neuropil labeling (*Kohl et al., 2014*).

We successfully identified the postsynaptic connections of 25 split-GAL4 lines (*Figure 1*, *Figure 1—figure supplements 1–23*, open access raw data video files are available at https://doi.org/10.26300/mttr-r782). *trans*-Tango signals from MB112C (MBON γ1pedc>α/β) and G0239 (MBON α3) were too weak and were excluded from further analysis. In contrast, signals from MB242A (MBON calyx) proved to be too noisy to confidently identify postsynaptic connections. We also employed three new split-GAL4 lines that had more specific expression for γ5β′2a, β′2mp, and α2sc MBONs. Postsynaptic connections of glutamatergic, GABAergic, and cholinergic MBONs vary with regard to the divergence and breadth of their postsynaptic connections (*Figure 1*, *Figure 1—figure supplements 1–23*, external open access raw data video files are available at https://doi.org/10.26300/mttr-r782). For instance, MB011B, which includes glutamatergic MBONs γ5β′2a, β′2mp, and β′2mp-bilateral has extensive connections across the superior protocerebrum (*Figure 1A*), whereas MB542B, which includes cholinergic MBONs α′1, α2p3p, α′3 m has limited connections within the LH (*Figure 1N*). The innervation patterns did not seem to correlate with neurotransmitter type or number of MBONs expressing each split-GAL4.

However, it was clear that some of the data were confounded by split-GAL4 lines that had off-target expression. We excluded extraneous signals by segmenting *trans*-Tango signals that were continuous with MBON terminals (*Figure 2A–B*) and then quantified the distribution of postsynaptic signals across brain regions in the standard brain (*Ito et al., 2014*). Nearly all MBONs have divergent connections across the dorsal brain regions, CRE, SMP, SIP, SLP, LH, as well as FSB, and LAL (*Figure 2C–E*, *Figure 2—figure supplement 1*).

### DANs are postsynaptic to MBONs

Of the DANs innervating the MB, 90% have dendritic arborizations that are localized to four of the five proposed MBON convergent regions, including CRE, SMP, SIP, and SLP (*Aso et al., 2014a*). Subsets of MBON axons overlapping with DAN dendritic arborizations provide feedback opportunities for MBONs to modulate DAN input thereby indirectly modulating MB circuits. Thus, we selected a subset of MBONs that were reported to co-localize with protocerebral anterior medial (PAM) DANs and co-stained with antibodies against tyrosine hydroxylase (TH) to identify overlap with *trans*-Tango signal (*Aso et al., 2014a*). As expected, some of the neurons postsynaptic to MBONs were TH positive; however, due to the complexity of *trans*-Tango-labeled neurons, we were unable to identify the DANs postsynaptic to a particular MBON unequivocally. Most overlap between TH and *trans*-Tango signals was observed with γ5β′2a (MB011B, 25 ± 0.7; n = 4; *Figure 3A*) and β′2mp

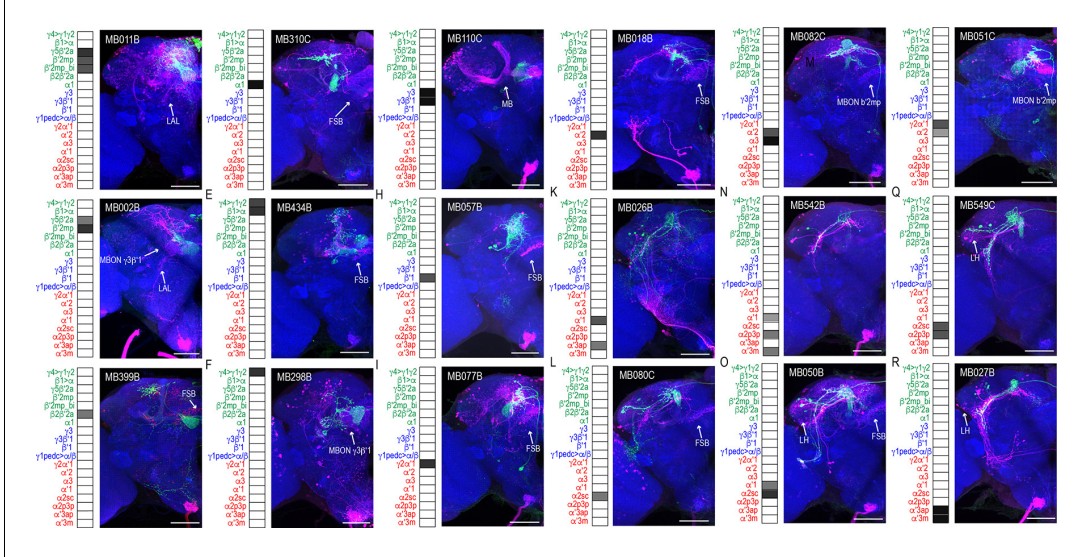

**Figure 1.** MBONs have divergent connections across the brain. Exemplar max-stacks of glutamatergic MBONs (**A**) MB011B, (**B**) MB002B, (**C**) MB399B, (**D**) MB310C, (**E**) MB434B, (**F**) MB298B, GABAergic MBONs (**G**) MB110C and (**H**) MB057B, and cholinergic MBONs (**I**) MB077B, (**J**) MB018B, (**K**) MB026B, (**L**) MB080C, (**M**) MB082C, (**N**) MB542B, (**O**) MB050B, (**P**) MB051C, (**Q**) MB549C and (**R**) MB027B, *trans*-Tango identified postsynaptic connections. For max-stacks: green, presynaptic MBONs, magenta, postsynaptic *trans*-Tango signal, blue, *brp-SNAP* neuropil. A map of the MBONs that are included in the expression pattern in each driver line accompanies each exemplar with the relative expression pattern (grayscale, 1–5) accordingly to FlyLight (https://splitgal4.janelia.org/cgi-bin/splitgal4.cgi). MBON maps are organized by neurotransmitter type: green=glutamatergic, blue=GABAergic, red=cholinergic. Scale bar = 50 μm.

The online version of this article includes the following figure supplement(s) for figure 1:

**Figure supplement 1.** MBON driver lines that have similar expression patterns also have similar postsynaptic connections across the brain.
**Figure supplement 2.** Full-size exemplar max-stack of MB011B.
**Figure supplement 3.** Full-size exemplar max-stack of MB002B.
**Figure supplement 4.** Full-size exemplar max-stack of MB399B.
**Figure supplement 5.** Full-size exemplar max-stack of MB310C.
**Figure supplement 6.** Full-size exemplar max-stack of MB434B.
**Figure supplement 7.** Full-size exemplar max-stack of MB298B.
**Figure supplement 8.** Full-size exemplar max-stack of MB110C.
**Figure supplement 9.** Full-size exemplar max-stack of MB057B.
**Figure supplement 10.** Full-size exemplar max-stack of MB077B.
**Figure supplement 11.** Full-size exemplar max-stack of MB018B.
**Figure supplement 12.** Full-size exemplar max-stack of MB026B.
**Figure supplement 13.** Full-size exemplar max-stack of MB080C.
**Figure supplement 14.** Full-size exemplar max-stack of MB082C.
**Figure supplement 15.** Full-size exemplar max-stack of MB051C.
**Figure supplement 16.** Full-size exemplar max-stack of MB549C.
**Figure supplement 17.** Full-size exemplar max-stack of MB027B.
**Figure supplement 18.** Full-size exemplar max-stack of MB074C.
**Figure supplement 19.** Full-size exemplar max-stack of MB210B.
**Figure supplement 20.** Full-size exemplar max-stack of MB433B.
**Figure supplement 21.** Full-size exemplar max-stack of MB083C.
**Figure supplement 22.** Full-size exemplar max-stack of MB051B.
**Figure supplement 23.** Full-size exemplar max-stack of MB077B.

(MB002B, 10.25 ± 1.3 n = 4 and MB074C, 4.75 ± 1.1, n = 4; *Figure 3B and C*) MBONs. These MBONs were predicted to co-localize with PAM DANs β′2p, β′2m and PAM DANs γ5 and β′2a, respectively (*Aso et al., 2014a*). Similarly, the γ3, γ3β′1 MBON was predicted to overlap with PAM γ3 and β′1m, and MB083C had an average of nine cells (9 ± 2.0, n = 10) with co-expression of TH and *trans*-Tango signals (*Figure 3D*). Likewise, the cholinergic γ2α′1 MBON (MB077C) was predicted to overlap with PAM γ4>γ1γ2, and indeed, MB077C brains averaged five cells (5 ± 1.5, n = 8) with

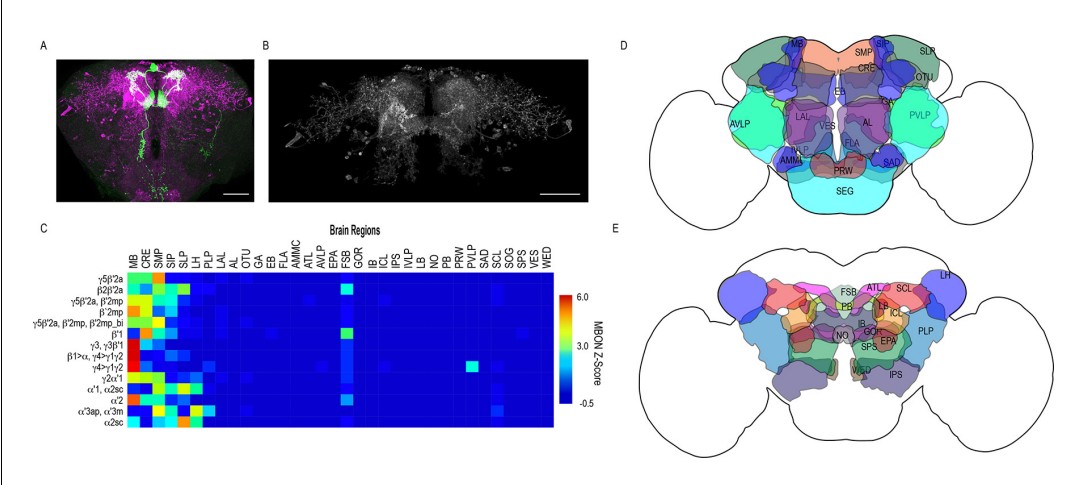

**Figure 2.** Whole brain distribution of MBON postsynaptic connections overlap. (**A**) Example of presynaptic MBON γ5β′2a (SS01308) and postsynaptic *trans*-Tango signal in a registered brain. For max-stacks: green, presynaptic MBONs, magenta, postsynaptic *trans*-Tango signal. (**B**) Example of segmented *trans*-tango signals that was continuous to MBON γ5β′2a terminals. For max-stack: gray, postsynaptic *trans*-Tango signal. (**C**) Heatmap displaying the overlap in segmented MBON postsynaptic signal by brain region. Postsynaptic signal for each MBON was normalized within each brain to capture respective expression levels. SS01308 was used to target MBON γ5β′2a, MB399B was used to target MBON β2β′2a, MB002B was used to target MBONs γ5β′2a, β′2mp, SS01143 was used to target MBON β′2mp, MB011B was used to target MBONs γ5β′2a, β′2mp, β′2mp_bi, MB057B was used to target MBON β′1, and MB110C was used to target MBONs γ3, γ3β′1. MB433B was used to target MBONs β1>α, γ4>γ1γ2, MB298B was used to target MBON γ4>γ1γ2, MB077C was used to target MBON γ2α′1 and MB50B was used to target MBONs α′1, α2sc. MB018B was used to target MBON α′2, MB027B was used to target MBON α′3ap, α′3 m, and SS01194 was used to target MBON α2sc. For raw postsynaptic signal see *Figure 2—figure supplement 1*. (**D**) Schematic of fly brain highlighting the most anterior brain regions included in mask analysis starting at AL and ending with SLP. (**E**) Schematic of fly brain highlighting the most posterior brain regions included in mask analysis starting at NO and ending with PB. AL: antennal lobe, AMMC: antennal mechanosensory and motor center, ATL: antler, AVLP: anterior ventrolateral protocerebrum, CRE: crepine, EB: ellipsoid body, EPA: epaulette, FSB: fan-shaped body, FLA: flange, GA: shoulder of lateral accessory lobe, GOR: gorget of ventral complex, IB: interior bridge, ICL: inferior clamp, IPS: inferior posterior slope, IVLP: inferior ventrolateral protocerebrum, LAL: lateral accessory lobe, LB: bulb of lateral complex, LH: lateral horn, MB: mushroom body, NO: noduli, OTU: optic tubercle, PB: protocerebral bridge, PLP: posterior lateral protocerebrum, PRW: prow, PVLP: posterior ventrolateral protocerebrum, SAD: saddle, SCL: superior clamp, SEG: subesophageal ganglion, SIP: superior intermediate protocerebrum, SLP: superior lateral protocerebrum, SMP: superior medial protocerebrum, SPS: superior posterior plate, VES: vest of ventral complex, WED: wedge. Scale bar = 50 μm.

The online version of this article includes the following figure supplement(s) for figure 2:

**Figure supplement 1.** Whole brain distribution of MBON postsynaptic connections.

co-expression of TH and *trans*-Tango signals per hemibrain (*Figure 3E*). There were a number of MBONs that had very few or no TH-positive postsynaptic neurons (*Figure 3—figure supplement 1*). The majority of these MBONs innervate the vertical lobe, including MBON α1 (MB310C; *Figure 3—figure supplement 1A*), MBON α′3ap, α′3 m (MB027B; *Figure 3—figure supplement 1E*), MBON α2sc (MB080C; *Figure 3—figure supplement 1F*) and MBON α′1, α2p3p, α′3m (MB542B; *Figure 3—figure supplement 1G*). MBON β′1 also had limited TH-positive postsynaptic neurons (MB057B; *Figure 3—figure supplement 1D*). Despite predictions that γ4>γ1γ2 MBON (MB298B) would co-localize with PAM γ4>γ1γ2, we found minimal co-expression of TH and *trans*-Tango signals (*Figure 3—figure supplement 1C*). This is likely a false negative due to the strength of the driver as annotations of the EM data has revealed postsynaptic connections with PAM γ4>γ1γ2 (*Clements et al., 2020*; *Li et al., 2020*). It is possible that the number of co-localized TH+ cells in our analysis here is an underestimation since some of the brains had fewer than expected TH+ neurons (*Figure 3—figure supplement 2*).

## Convergent MBONs

Whole brain overlap analysis identified the MB itself as a site of rich convergence for most MBON lines (*Figure 2C*). MBON postsynaptic signals in MB were not surprising given that many MBONs provide feedforward connections between MB compartments (*Aso et al., 2014a*). For instance, MBON γ4>γ1γ2 has dendritic arbors in γ4 and axonal projections in γ1γ2, MBON γ1pedc>α/β have

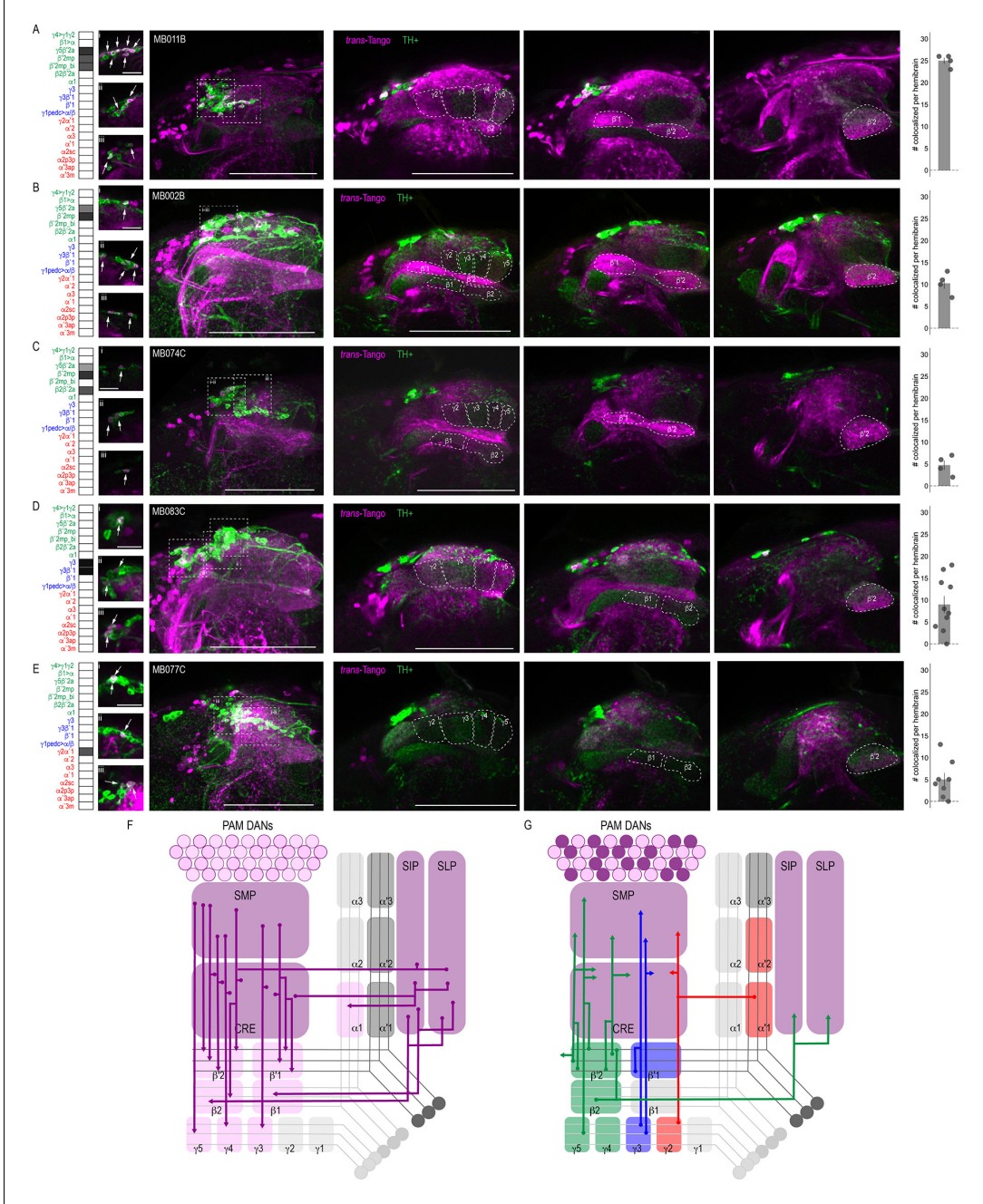

**Figure 3.** DANs postsynaptic to MBONs. Exemplar max-stacks of MBON lines in which TH+ cells overlapped with postsynaptic signal of glutamatergic (A) MBON γ5β′2a, β′2mp, β′2mp_bilateral (MB011B), (B) MBON γ5β′2a, β′2mp (MB002B), (C) MBON γ5β′2a, β′2mp, β2β′2a (MB074C), (D) GABAergic MBONs γ3, γ3β′1 (MB083C) and (E) cholinergic MBONs γ2α′1 (MB077C). Overlapping TH+ and *trans*-Tango cell bodies are highlighted in insets, scale bar = 10 μm. Max stacks of MB are included (Column I), scale bar = 50 μm. Column II-IV depict single optical planes from anterior to posterior outlining MB compartments. Bar graphs indicate the average number of co-localized cells per hemibrain (mean +/- standard error). Green, TH-positive cells; magenta, postsynaptic *trans*-Tango signal. MBON maps are organized by neurotransmitter type: green=glutamatergic, blue=GABAergic, red=cholinergic. (F) Schematic depicting the MB innervation by PAM DANs. PAM DANs extend dendrites to SMP, CRE, SIP, and SLP. (G) Schematic depicting the MBONs that synapse on TH+ cells.

The online version of this article includes the following figure supplement(s) for figure 3:

**Figure supplement 1.** DANs postsynaptic to MBONs.
**Figure supplement 2.** Total PAM TH+ cells counted.

dendritic arbors in γ1 and axonal projections in α/β lobes, and MBON β1>α has dendritic arbors in β1 and axon projections to the entire alpha lobe. However, further analysis revealed that in addition to providing connections between MB compartments, MBONs converge directly on other MBONs presumably through axo-axonal connections. Two different MBONs are frequently targeted: MBON β′2mp (*Figure 4A*) and MBON γ3β′1 (*Figure 4B*). Interestingly, MBON β′2mp receives convergent glutamatergic, GABAergic, and cholinergic input from MBON γ5β′2a (MB011B and MB210B), MBON γ3β′1 (MB110C and MB83C), MBON α′2 (MB018B and MB082C), and MBON γ2α′1 (MB077B and MB051C) (*Figure 4A*, *Figure 4—figure supplement 1*). MBON γ3β′1 receives convergent input from glutamatergic MBON β′2mp as revealed with split-GAL4 lines MB002B (*Figure 4B*) and MB074C (*Figure 4—figure supplement 1*) and glutamatergic MBON γ4>γ1γ2 (MB298B, *Figure 4B*). We hypothesize that similar to MBONs that project to other regions of the MB, MBON γ3β′1, and MBON β′2mp create opportunities for multilevel feedforward networks to update information to drive behavioral response (*Figure 4C*).

## Convergence outside the MB

Another site of convergence of the MBON network was the FSB (*Figure 5*). MBON postsynaptic connections display a laminar organization primarily across the dorsal region of the FSB. Nearly all the glutamatergic and GABAergic MBONs converge onto FSB layers 4 and 5, and to a lesser extent, layer 6 (*Figure 5A–B*). MBON α1 is the only type of MBON that had broad *trans*-Tango signals in the FSB (*Figure 5A*). To rule out sexual dimorphism in the postsynaptic connections of MBON α1,

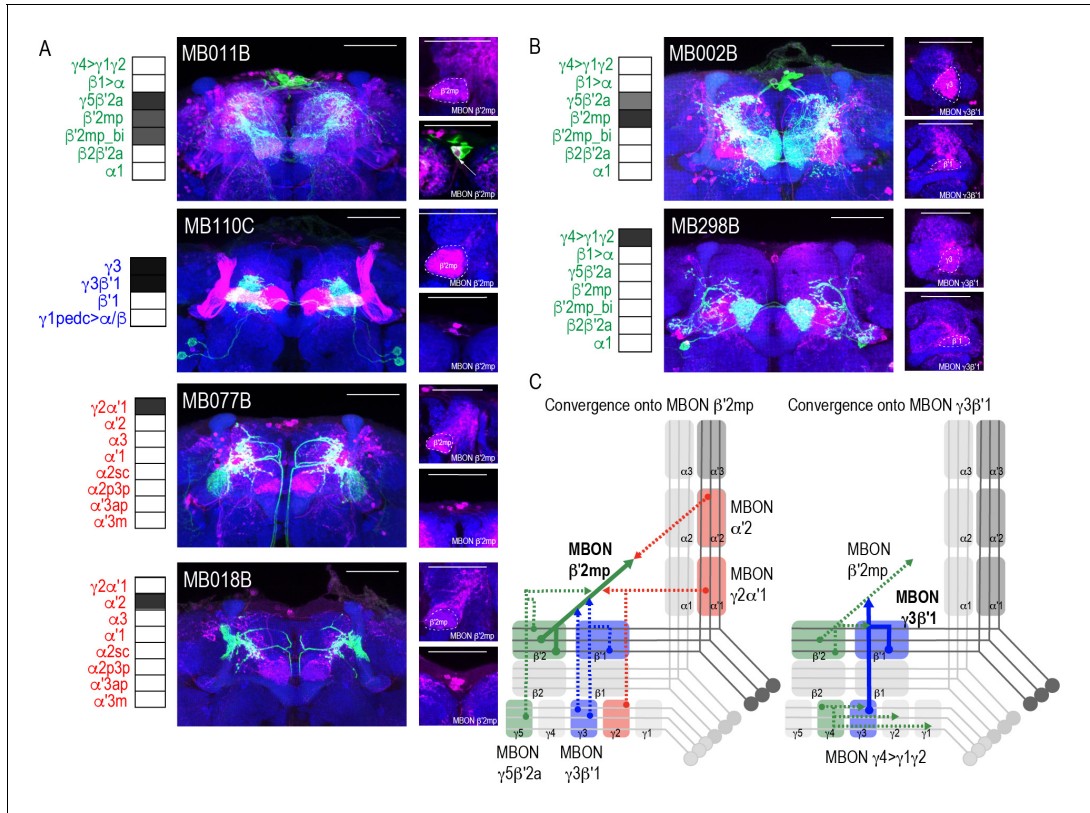

**Figure 4.** Subsets of MBONs converge on other MBONs. (**A**) MBON β′2mp receives convergent input from glutamatergic MBON γ5β′2a (MB011B), GABAergic MBONs γ3, γ3β′1 (MB110C) and cholinergic MBON γ2α′1 (MB077B) and MBON α′2 (MB018B). (**B**) MBON γ3β′1 receives convergent input from glutamatergic MBON β′2mp (MB002B) and MBON γ4>γ1γ2 (MB298B). β′2mp, γ3 and β′1 are outlined in representative stacks. (**C**) Schematics summarizing identified convergent MBONs (β′2mp and γ3β′1) and their respective convergent input. Solid lines represent the convergent MBON and dotted lines represent convergent input. For max-stacks: green, presynaptic MBONs, magenta, postsynaptic *trans*-Tango signal, blue, *brp-SNAP* neuropil, scale bar=50 μm.

The online version of this article includes the following figure supplement(s) for figure 4:

**Figure supplement 1.** Patterns of MBON convergence is consistent across MBON driver lines that have similar MBON expression.

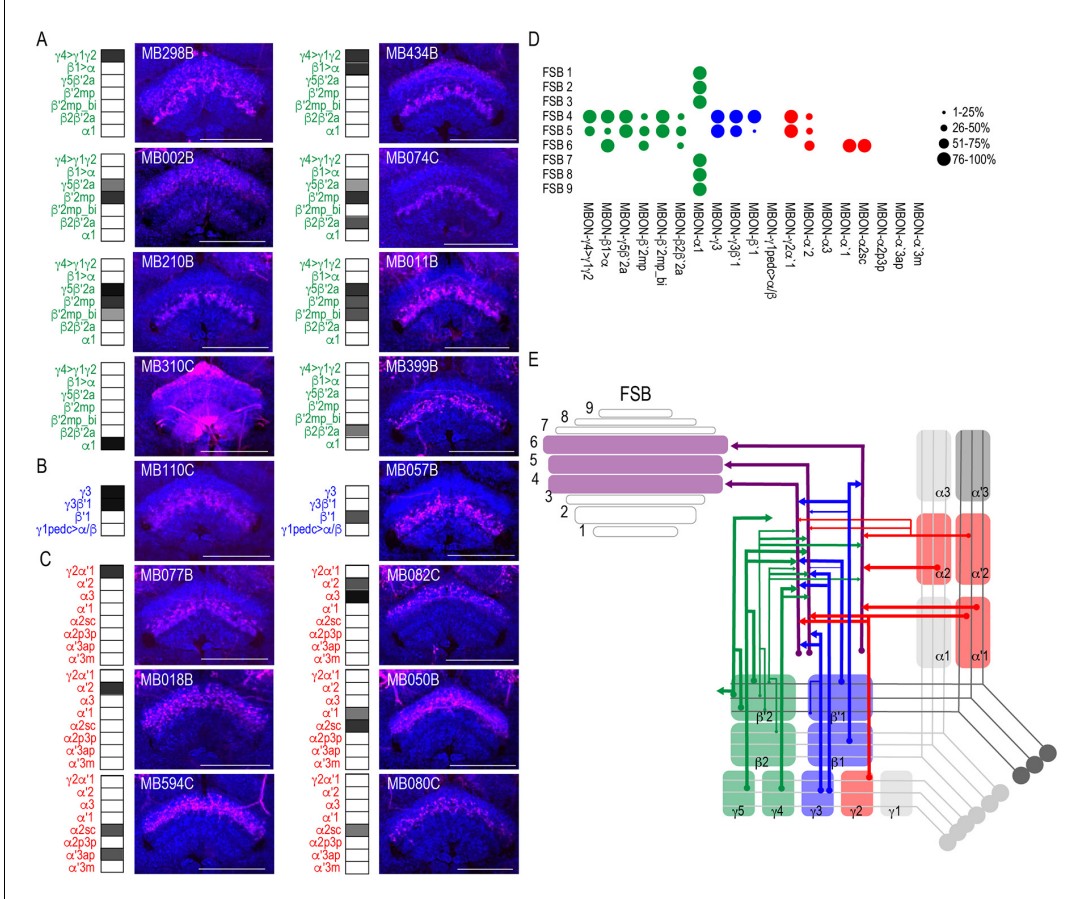

**Figure 5.** MBONs converge on different layers of the FSB. Exemplar max-stacks of glutamatergic (A), GABAergic (B), and cholinergic (C) MBONs whose postsynaptic neurons innervate the FSB. Max-stacks are approximately 50 μm thick. Slices were selected based on the relative position of the FSB. For FSB stacks: magenta, postsynaptic *trans*-Tango signal, blue, *brp-SNAP* neuropil. Map of MBONs accompany each exemplar with the relative expression pattern (grayscale, 1–5) accordingly to FlyLight. For each map, green=glutamatergic, blue=GABAergic, red=cholinergic. Scale bar = 50 μm. (D) Map summarizing the percentage of *trans*-Tango-positive signal in each FSB layer across brains for each MBON. (E) Schematic depicting MBONs that converge onto different layers of the FSB. MB compartments are colorized based on the neurotransmitter expressed by the MBON that innervates it. Lines thickness corresponds to the percentage of *trans*-Tango-positive signal in each FSB layer across brains for each MBON.

The online version of this article includes the following figure supplement(s) for figure 5:

**Figure supplement 1.** MBON α1 postsynaptic signal innervating FSB in females.

**Figure supplement 2.** Variability in FSB postsynaptic signal.

we compared *trans*-Tango signal in the FSB in male and female brains and found similar innervation patterns (**Figure 5—figure supplement 1**). Cholinergic MBONs also had *trans*-Tango signals in the dorsal FSB but with more variability across MBON lines and within each line (**Figure 5C**). For instance, *trans*-Tango with MBON γ2α′1 consistently visualized projections to FSB layers 4 and 5 in all the brains analyzed, whereas more variability was observed in FSB innervation pattern across MBON α′2 brains (**Figure 5—figure supplement 2**). MBON α′1 and α2sc both project exclusively to FSB layer 6 (**Figure 5C**). Together, FSB layers 4 and 5 receive convergent input from combinations of glutamatergic, GABAergic and cholinergic MBONs (**Figure 5D**; **Figure 5E**).

Both visual and computational analyses confirmed the CRE, SMP, SIP, and SLP, as well as the MB and FSB as obvious postsynaptic targets of the MBON network. Visual inspection also confirmed the LAL as postsynaptic to multiple MBON lines. Its identification was less obvious in computational analysis largely because the neurites innervating the LAL were not as extensive as the LAL itself and were often difficult to segment. Although not extensive, LAL innervation was consistent across glutamatergic, GABAergic, and cholinergic MBONs (**Figure 6**). Specifically, glutamatergic γ5β′2a, β′2mp, and β′2mp_bilateral had postsynaptic neurites within the LAL in all of the brains analyzed

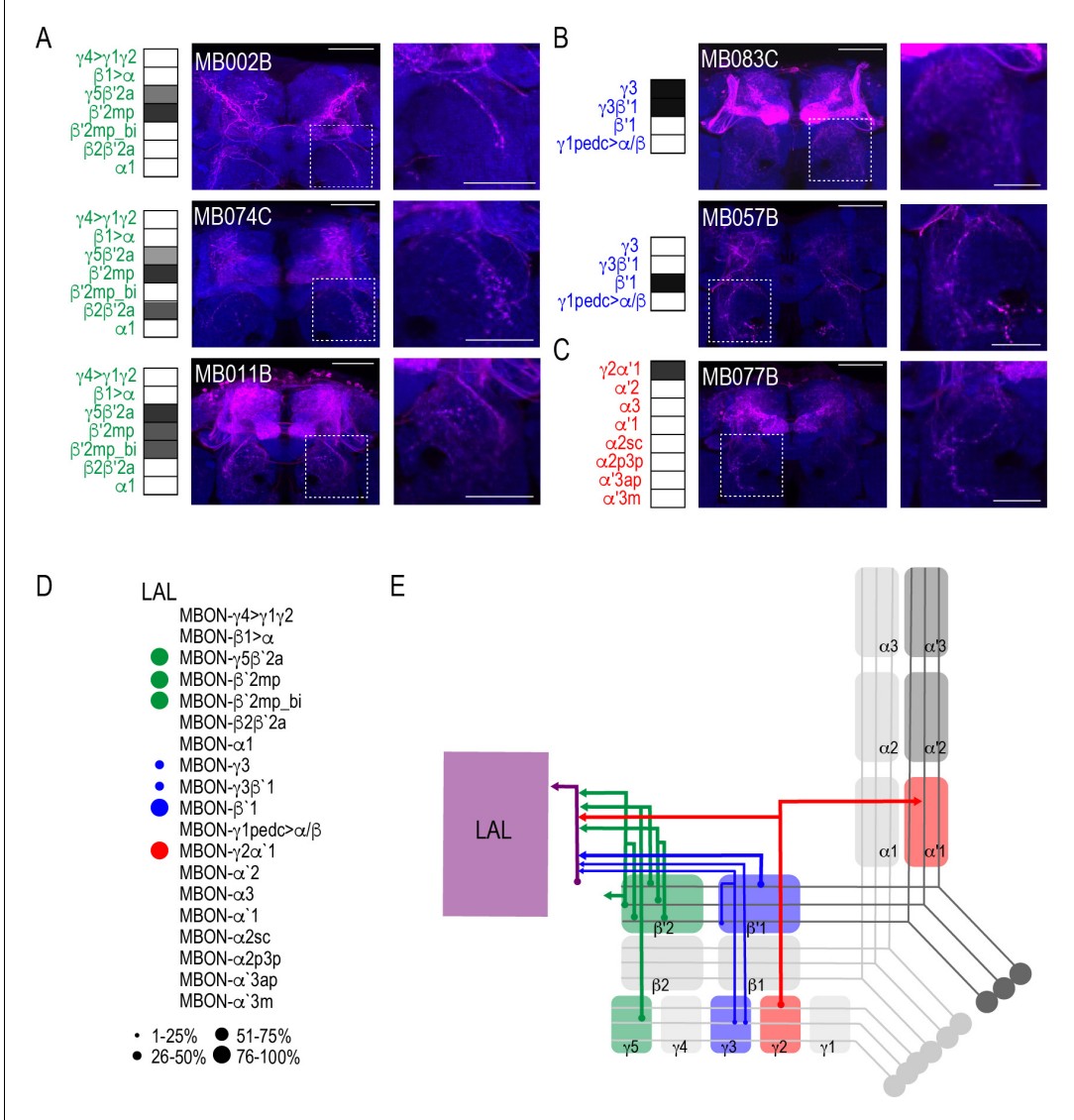

**Figure 6.** MBONs converge onto LAL neurons. Exemplar max-stacks of glutamatergic (**A**), GABAergic (**B**), and cholinergic (**C**) MBONs innervating the LAL. Max-stacks are approximately 50 μm thick. Slices were selected based on the relative position of the LAL. Magenta, postsynaptic *trans*-Tango signal, blue, *brp-SNAP* neuropil. Map of MBONs accompany each exemplar with the relative expression pattern (grayscale, 1–5) accordingly to FlyLight. For each map green=glutamatergic, blue=GABAergic, red=cholinergic. Scale bar = 50 μm. Scale bar for insets = 10 μm (**D**) Map summarizing the percentage of *trans*-Tango-positive signal in LAL across brains for each MBON. (**E**) Schematic depicting MBONs that converge onto neurons innervating the LAL. MB compartments are colorized based on the neurotransmitter expressed by the MBON that innervates it. Lines thickness corresponds to the percentage of *trans*-Tango-positive signal in LAL across brains for each MBON.

(*Figure 6A*). Similarly, GABAergic MBON γ3, γ3β′1, and β′1 (*Figure 6B*) and cholinergic MBON γ2α′1 (*Figure 6C*) consistently had postsynaptic neurites within the LAL. Thus, like the FSB, neurons innervating the LAL receives convergent input from combinations of glutamatergic, GABAergic and cholinergic MBONs (*Figure 6D*; *Figure 6E*).

Thus far, we have confirmed two postsynaptic targets of the MBON network that reside outside of the MB: the FSB and LAL. However, the identities of the postsynaptic neurons within FSB and LAL as well as their functions remain unknown. Our strategy for identifying FSB and LAL neurons and interrogating their functional connectivity with MBONs was to selectively label neurons in FSB and LAL using specific drivers and to examine whether they are co-localized with postsynaptic signal when we initiate *trans*-Tango from MBONs. To achieve this, we identified candidate FSB and LAL LexA lines by performing a mask search of the LexA lines that have overlapping expression within

the convergent region and brought them together with MBON lines: MB051C and MB077C were used to target MBON γ2α′1, MB083C and MB110C were used to target γ3β′1, and MB074C was used to target MBON β′2mp. We identified three candidate LexA lines: one to target FSB layer four neurons - R47H09 (*Jenett et al., 2012*; *Pfeiffer et al., 2013*; *Pfeiffer et al., 2010*), and two to target LAL neurons - VT055139 and VT018476 (*Tirian and Dickson, 2017*). Finally, we generated *trans*-Tango reporter flies where the UAS-myrGFP was replaced with UAS-CD2, and LexAOp-mCD8::GFP was included in order to visualize the starter MBONs, the postsynaptic *trans*-Tango signal, and the LexA lines simultaneously.

We successfully combined the majority of the targeted MBON split-Gal4 lines with FSB and LAL LexA lines (we were unable to combine MB074C with LexA line 47H09). Interestingly, for the cholinergic MBON γ2α′1 (MB077C), we identified at least two postsynaptic FSB neurons (labeled in the 47H09 LexA line; *Figure 7A*) and at least five postsynaptic LAL neurons (labeled in the VT055139 LexA line; *Figure 7B*). We next sought to interrogate functional connectivity between MBON γ2α′1 and 47H09 FSB neurons and VT055139 LAL neurons by combining optogenetic stimulation of MBON γ2α′1 using UAS-Chrimson and functional calcium imaging in FSB and LAL using LexAop-GCaMP6s. Stimulation of cholinergic MB077C with 400–500 ms of red light (627 nm) resulted in an increase in calcium signal in the FSB and LAL (*Figure 7C*). Similar activation of other cholinergic MBONs (MB080C), which do not innervate the LAL or layer 4 of the FSB, did not result in signal (*Figure 7—figure supplement 1*), supporting the specificity of this interaction and suggesting that the MBON γ2α′1 is both anatomically and functionally connected to the FSB and LAL. Strikingly, GABAergic MBON γ3β′1 (MB083C) also had at least one identified postsynaptic FSB neuron that was included in the expression of FSB 47H09 LexA line (*Figure 7D*) and at least two identified postsynaptic LAL neurons that were included in the expression of LAL VT055139 LexA line (*Figure 7E*). Thus, the genetically identified subsets of LAL and FSB neurons receive convergent input from GABAergic and cholinergic MBONs (*Figure 7F*). We hypothesize that the convergence of excitatory and inhibitory input onto both the LAL and FSB is critical for guiding behavior.

Finally, to determine the role of LAL neurons in the context of guiding behavior of flies in groups, we performed analyses of group activity using thermogenetic inactivation of identified split-GAL4 LAL neurons (*Scaplen et al., 2019*). Individual flies were tracked offline using Flytracker to obtain activity-based features (*Eyjolfsdottir et al., 2014*). Inactivation of SS32219-GAL4-positive LAL neurons (*Figure 7G*) resulted in significant increases in group activity (*Figure 7H*, $F_{(2,21)}=39.28$ $p<0.0001$), pathlength ($F_{(2,29)}=33.39$, $p<0.0001$), angular velocity ($F_{(2,29)}=51.87$, $p<0.0001$) and velocity ($F_{(2,29)}=30.97$, $p<0.0001$) of individual flies (*Figure 7I*)). Behavioral results were replicated with a separate LAL split-GAL4 line (SS32230-GAL4, *Figure 7—figure supplement 2*), suggesting that LAL neurons downstream of MBONs modulate locomotor activity of flies in a group. Group activity at permissive temperatures was not different from controls (*Figure 7—figure supplement 3*).

## Discussion

The MB is a high-level integration center in the *Drosophila* brain with an established role in learning and memory. The iterative nature of converging and diverging MB neural circuits provides an excellent example of the anatomical framework necessary for complex information processing. For instance, on a rapid timescale, interactions between MB compartments could generate different output patterns to drive behavior, whereas on a slower timescale, interactions between MB compartments could reevaluate memories of a context (*Aso and Rubin, 2016*; *Felsenberg et al., 2017*; *Felsenberg et al., 2018*).

We sought to map the projections from the MB using the genetic anterograde transsynaptic technique, *trans*-Tango. We report the connectivity of MBONs across multiple subjects in both males and females and highlight the variability in connectivity that potentially exists across animals. Our study complements the ongoing efforts of EM reconstruction of a whole brain of a single female fruit fly and confirms previous anatomical predictions (*Aso et al., 2014a*). Although the complete EM dataset of an adult fly brain has been an invaluable resource that significantly accelerated the mapping of the neural circuits underlying innate and learned behaviors (*Adden et al., 2020*; *Li et al., 2020*; *Xu et al., 2020*; *Zheng et al., 2018*), the massive undertaking of acquiring a full EM dataset renders it impractical to perform for multiple individuals. Thus, *trans*-Tango, expands the value of

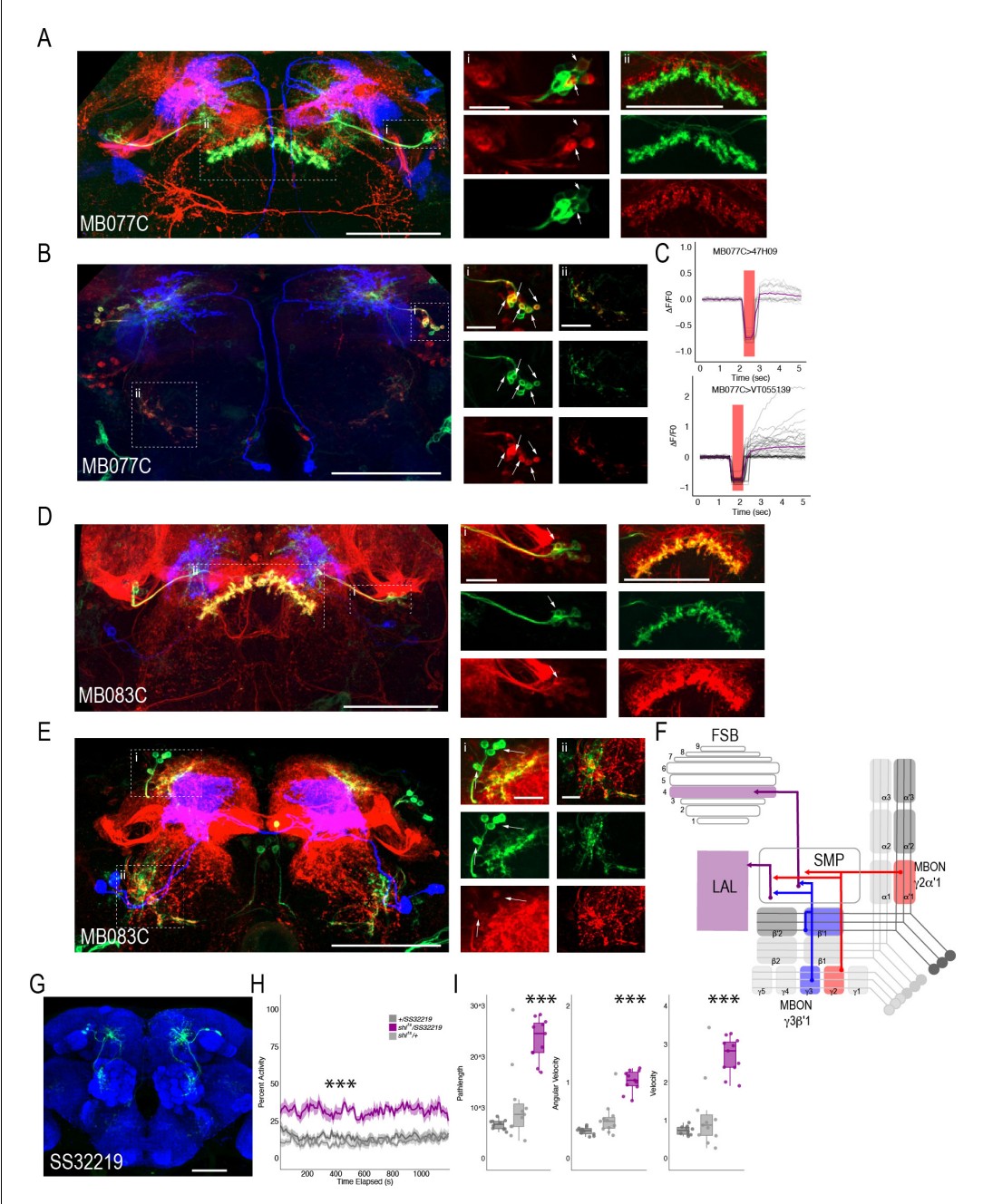

**Figure 7.** MBONs γ3β'1 and γ2α'1 converge onto the same subset of LAL and FSB neurons. Exemplar max-stacks of cholinergic MBON γ2α'1 (MB077C) postsynaptic connections and identified overlap with respective (**A**) FSB (47H09) and (**B**) LAL (VT015539). (**C**) Confirmation of functional connection with optogenetic activation of MB077C and calcium imaging of FSB neurons in SMP and FSB (47H09), and calcium imaging of LAL neurons in SMP (VT015539). The red bar indicates when the LED was on and the shutter was closed to protect the PMTs during LED stimulation. Exemplar max-stacks of GABAergic MBON γ3β'1 (MB083C) postsynaptic connections and identified overlap with respective (**D**) FSB (47H09) and (**E**) LAL (VT015539). Max-stacks are approximately 50 µm thick. Slices were selected based on the relative position of the LAL and FSB. In A, B, D and E, red, postsynaptic *trans*-Tango signal; blue, CD2 marker of split-GAL4 line; green, LexA FSB or LAL. Scale bar = 50 µm. (**F**) Schematic highlighting convergence of MBONs γ3β'1 and γ2α'1 onto the same genetically identified subsets of LAL and FSB neurons. (**G**) Max-stack of SS32219; green, GFP expression; blue, neuropil. Scale bar = 50 µm. (**H**) *shibire*^ts (*shi*^ts) inactivation of LAL using split-GAL4 SS32219 resulted in significant increases in group activity ($F_{(2,21)}=39.28$ p<0.0001). Group activity counts were binned over 10 s periods, averaged across biological replicates of 10 flies each (n = 8) and plotted against time. Lines depict mean +/- standard error. (**I**) One video was selected at random of each genotype and processed using FlyTracker to calculate the average pathlength ($F_{(2,29)}=33.39$, p<0.0001), angular velocity ($F_{(2,29)}=51.87$, p<0.0001) and velocity ($F_{(2,29)}=30.97$, p<0.0001) of individual flies. Box plots with

*Figure 7 continued on next page*

*Figure 7 continued*

overlaid raw data were generated using RStudio. Each dot is a single fly. One-way ANOVA with Tukey Posthoc was used to compare mean and variance. ***p<0.0001.

The online version of this article includes the following figure supplement(s) for figure 7:

**Figure supplement 1.** Optogenetic activation of MBON α2sc (MB080C) does not result in changes in signal recorded from of LAL neurons in SMP (VT015539).

**Figure supplement 2.** Inactivation of LAL using split-GAL4 SS32230 results in significant increases in group activity.

**Figure supplement 3.** Group activity of split-GAL4 SS32219 and SS32230 at permission temperatures.

the EM reconstruction data by examining circuit connectivity across multiple individuals. Further, *trans*-Tango can be readily adapted to functional studies in which the activity of the postsynaptic neurons is altered by expressing optogenetic/thermogenetic effectors or monitored by expressing genetically encoded sensors. Our tracing studies reported here serve as the foundation for these future experiments.

Our studies reveal that the MB circuits are highly interconnected with multiple regions of converging projections both within and downstream of the MB. Our experiments also show diverging projections in the downstream postsynaptic targets. We identify, both anatomically and functionally, a multilayer circuit that includes GABAergic and cholinergic MBONs that converge on the same subset of FSB and LAL neurons. This circuit architecture allows for rapid updating of the online processing of sensory information before executing behavior. Further, this circuit organization is likely a conserved motif among insects (*Strausfeld and Hirth, 2013a*; *Strausfeld et al., 2009*; *Strausfeld et al., 2020*; *Wolff and Strausfeld, 2015a*).

## Anatomical divergence across the brain

Successive levels of convergence and divergence across the brain permit functional flexibility (*Jeanne and Wilson, 2015*; *Man et al., 2013*; *Tye, 2018*). Like the mushroom body, cerebellar circuits in mammals exhibit large divergence in connectivity, and this can support diverse types of synaptic plasticity (*Litwin-Kumar et al., 2017*). Previous neuroanatomical work in insects described divergent afferent and efferent MB neurons, although the extent of this divergence was unknown (*Ito et al., 1998*; *Li and Strausfeld, 1997*; *Li and Strausfeld, 1999*; *Mao and Davis, 2009*; *Nässel and Elekes, 1992*; *Tanaka et al., 2008*; *Waddell, 2013*). Our data revealed varying levels of divergence of postsynaptic connections of MBONs across the brain. Every one of the analyzed MBONs had postsynaptic partners projecting to multiple brain regions (*Figure 2C*, *Figure 8A*). Further, nearly the entire superior protocerebrum as well as portions of the inferior protocerebrum received input from at least one MBON, providing opportunities for comprehensive integration of signals from the MBON network.

## Convergence within MBONs

Multiple feedforward and feedback circuits exist within the MB (*Aso et al., 2014a*; *Eichler et al., 2017*; *Eschbach et al., 2020*; *Otto et al., 2020*; *Takemura et al., 2017*; *Zheng et al., 2018*). Our data revealed at least two MBONs that receive convergent input from multiple MBONs and are also reciprocally connected (*Figure 3*, *Figure 8B*). The convergent MBON input to β′2mp is especially interesting as cholinergic (MBON γ2α′1), GABAergic (MBON γ3β′1), and glutamatergic (MBON γ5β′2a) MBONs drive opposing behaviors (*Aso et al., 2014b*). For instance, activation of the cholinergic or GABAergic MBON results in naive odor preference, whereas activation of the glutamatergic MBON results in robust naive avoidance (*Aso et al., 2014b*; *Lewis et al., 2015*). Similarly, the cholinergic MBON activity mediates aversive associations (*Berry et al., 2018*; *Yamazaki et al., 2018*), whereas glutamatergic MBON activity mediates appetitive associations and extinction of aversive memories (*Felsenberg et al., 2018*; *Owald et al., 2015*; *Yamazaki et al., 2018*).

Considering that MBON β′2mp receives convergent input from these parallel and opposing pathways (*Figure 4C*), it likely serves as a decision hub by integrating activity to modulate cue-induced approach and avoidance behavior. How MBON β′2mp integrates information across MBONs and drives behavioral responses remains to be determined. Naïve activation of MBON β′2mp does not appear to influence behavioral choice, it instead acts as a sleep suppressor (*Aso et al., 2014b*).

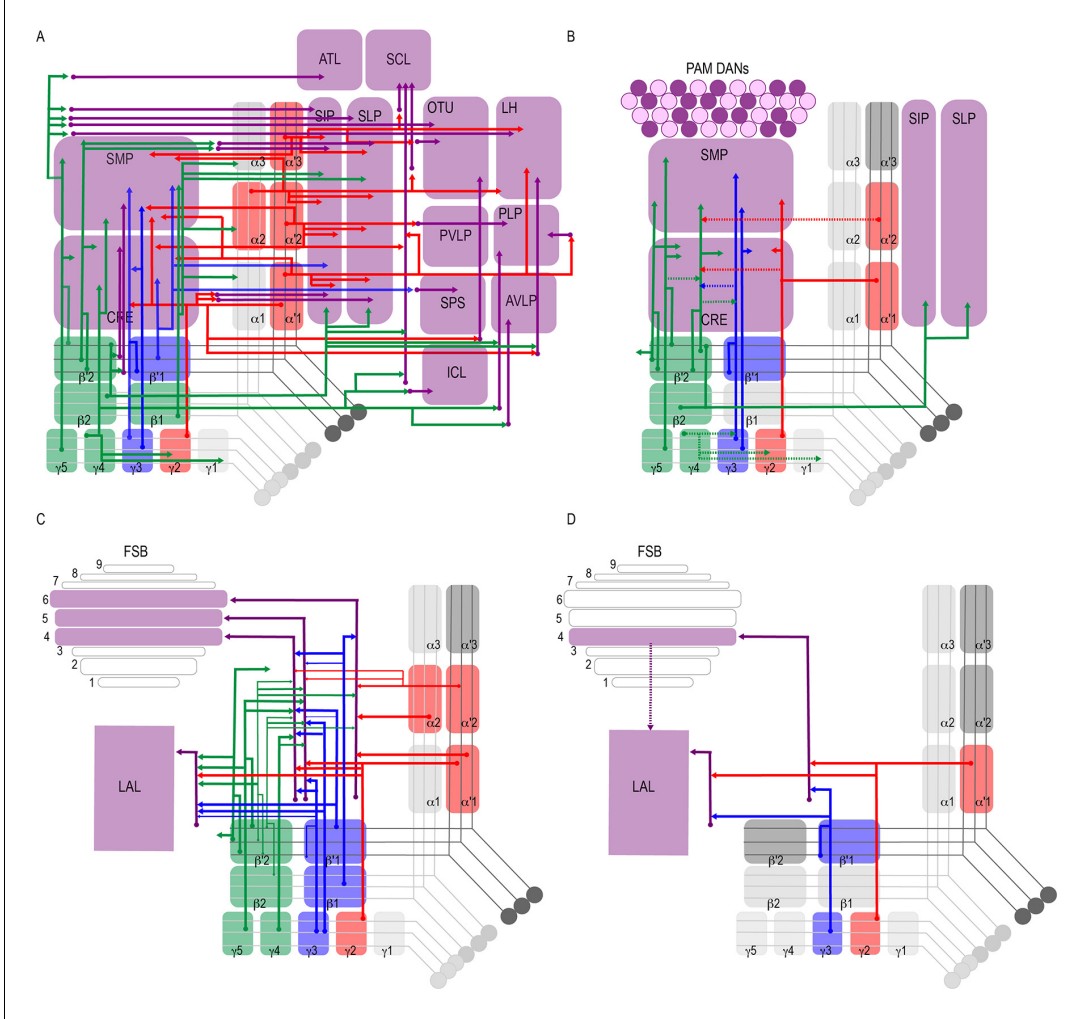

**Figure 8.** Summary schematics highlighting postsynaptic connections of MBON innervating (**A**) innervating the protocerebrum (**B**) PAM DANs (solid lines) and MBONs (dotted lines). (**C**) FSB and LAL. Lines thickness corresponds to the percentage of *trans*-Tango-positive signal in FSB and LAL across brains for each MBON. (**D**) Schematic highlighting convergence of MBONs γ3β′1 and γ2α′1 onto the same genetically identified subsets of LAL and FSB neurons (solid lines). Dotted lines depict the established connections between the FSB and LAL (***Wolff and Strausfeld, 2015b***).

Inhibition of MBON β′2mp during sleep enhances long-term memory (***Scaplen et al., 2020***). Separately, local protein synthesis within MBON β′2mp, has been implicated in the consolidation of long-term memory (***Wu et al., 2017***). This makes MBON β′2mp an ideal model for understanding how sleep and memory signals might be integrated at a molecular level. It should be mentioned that MBON γ3β′1 reportedly acts as a sleep activator (***Aso et al., 2014b***) and local protein synthesis within this MBON is also important for the consolidation of long-term memory (***Wu et al., 2017***). Thus, MBON γ3β′one likely also plays a role in integrating sleep and memory signals through its reciprocal connections MBON β′2mp.

This provides a well-characterized anatomical framework to understand how opposing memories are acquired, consolidated, expressed and updated. Since the roles of these converging MBONs in naive and learned behaviors are state dependent (***Grunwald Kadow, 2019***; ***Lewis et al., 2015***; ***Tsao et al., 2018***), we hypothesize that MBON γ3β′1 and MBON β′2mp, both receiving convergent input from other MBONs, providing opportunities for feedforward networks to update information processing depending on the state of the animal.

## Convergence within DANs

Some of the feedback connections originally hypothesized to exist in the MB were between MBONs and DANs (*Aso et al., 2014b*; *Ichinose et al., 2015*). Our analysis revealed neurons postsynaptic to MBONs that are TH positive (*Figure 3*, *Figure 8B*). Recent studies that combined EM annotation and calcium imaging to identify specific MBON-DAN connections suggest extensive recurrent connectivity between MBONs and DANs, validating our findings (*Felsenberg et al., 2018*; *Li et al., 2020*; *Otto et al., 2020*). For example, previous studies using both GFP Reconstitution Across Synaptic Partners (GRASP) and EM annotation revealed that MBON α1 and DAN α1 are synaptically connected (*Ichinose et al., 2015*; *Li et al., 2020*). We similarly identified a few DAN neurons that innervate the horizontal MB lobes within the MBON α1 postsynaptic signal. A recent study showed that the 20 DANs that innervate the γ5 MB compartment are clustered into five different subtypes that innervate distinct anatomical regions within the γ5 compartment (*Otto et al., 2020*). According to this study, only one of the γ5 DANs receives direct recurrent feedback from γ5β′2a MBONs (*Otto et al., 2020*). Based on these recent anatomical characterizations, we believe that the TH+ neurons within the postsynaptic signal of γ5β′2a are the γ5 DANs.

## Convergence within the FSB

The FSB is the largest substructure of the central complex, and it serves as a sensory-motor integration center (*Pfeiffer and Homberg, 2014*; *Wolff et al., 2015*). The FSB comprises nine horizontal layers (*Wolff et al., 2015*) that are innervated by large-field neurons (*Hanesch et al., 1989*). Previous work in blow flies (*Phillips-Portillo and Strausfeld, 2012*) and, later work in *Drosophila* (*Aso et al., 2014a*), predicted that the FSB was postsynaptic to output neurons of the MB. Our data confirm that the large-field, tangential neurons of the dorsal FSB are postsynaptic to the majority of MBONs. Although there exists some variation across brains (*Figure 5—figure supplement 2*), glutamatergic and GABAergic MBONs predominately project to FSB layers 4 and 5, whereas cholinergic MBONs mainly project to FSB layer 6. Connections between MBONs and FSB were consistent across different split-GAL4 lines that have overlapping expression patterns. Similar extensive direct connectivity between these MBONs and the dorsal FSB, especially layers 4 and 5, were found in the recently annotated EM hemibrain dataset (*Li et al., 2020*). Together, these observations suggest that the connectivity between the MB and FSB are structurally, and perhaps in some cases functionally, conserved across insect species.

How are FSB layers 4/5 and 6 functionally distinct? The dorsal FSB has a well-established role in modulating sleep and arousal (*Berry et al., 2015*; *Donlea et al., 2011*; *Ueno et al., 2012*), locomotor control (*Strauss, 2002*), courtship (*Sakai and Kitamoto, 2006*), and visual memory (*Li et al., 2009*; *Liu et al., 2006*; *Wang et al., 2008*). FSB layer 5 has been specifically implicated in processing information regarding elevation in a *foraging*- and *rutabaga*-dependent manner (*Li et al., 2009*). More recent studies have implicated the dorsal FSB in processing nociceptive information (*Hu et al., 2018*). FSB layer 6 plays a specific role in avoidance of a conditioned odor, whereas layers 4 and 5 respond to aversive stimuli and are responsible for innate, but not conditioned, avoidance (*Hu et al., 2018*). Moreover, recent connectome data suggest that differences exist in the postsynaptic connections of layers 4/5 and 6 as well. Overall, there is high degree of interconnectivity within the FSB (*Clements et al., 2020*). The predominate output of FSB layer 6 neurons are other FSB neurons. In fact, many FSB layer 6 neurons project exclusively to other FSB neurons (*Clements et al., 2020*). In contrast, FSB layer 4 neurons send direct projections to other brain structures - CRE, SMP, and LAL - in addition to projecting to other FSB neurons. The connections with the LAL position the FSB layer 4 to directly influence downstream motor output signals prior to executing behavior. Recent EM analysis also suggests that some FSB layer 6 neurons synapse back onto PAM DAN neurons (*Li et al., 2020*). This connectivity is in line with the associative role in conditioned nociception avoidance described for FSB layer 6 (*Hu et al., 2018*).

Interestingly, we found that the pattern of FSB postsynaptic targets of the MBONα1 is dissimilar to other glutamatergic MBONs. FSB layers 4/5 and 6 are not present in the MBON α1 postsynaptic signal. Instead, MBON α1 project to neurons that innervate the ventral and most dorsal aspect of the FSB. The ventral FSB is implicated in innate avoidance of electric shock (*Hu et al., 2018*), and more recent data suggest that its activity is tuned to airflow cues for orientation during flight (*Currier et al., 2020*). Artificial activation of MBON α1 does not result in significant avoidance

behavior (*Aso et al., 2014b*). However, it has been implicated in the acquisition, consolidation, and expression of 24 hr long-term sucrose memory (*Ichinose et al., 2015*). It is possible that MBON α1 provides appetitive valence signals to the ventral FSB to guide goal-directed flight. Functionally validating the role of MBON α1 and its relationship with its putative downstream neurons is key to appreciating how learning signals can drive behavioral decisions.

More research is necessary to further understand the functional role of different FSB layers and how information is integrated across these layers. Based on the anatomical data, it is clear that although the MB and FSB can function in parallel during memory formation, they act as parts of a dynamic system to integrate information and adjust behavioral responses.

## Convergence within the LAL

The LAL is an important premotor waystation for information traveling from the central complex to descending neurons innervating thoracic motor centers across insects (*Chiang et al., 2011*; *Franconville et al., 2018*; *Hanesch et al., 1989*; *Wolff and Strausfeld, 2015a*; *Wolff and Strausfeld, 2015b*). Accordingly, the LAL has been implicated in orientation to pheromones in the moth (*Kanzaki et al., 1991a*; *Kanzaki et al., 1991b*; *Mishima and Kanzaki, 1999*; *Namiki et al., 2014*; *Namiki et al., 2018*; *Wada and Kanzaki, 2005*), flight in the locust and dragonfly (*Homberg, 1994*; *Olberg, 1986*), locomotion in *Drosophila* (*Bidaye et al., 2014*) stimulus-directed steering in *Drosophila,* the cockroach, cricket, and moth (*Harley and Ritzmann, 2010*; *Rayshubskiy et al., 2020*; *StaudacherY, 1998*; *Zorović and Hedwig, 2011*) and in response to mechanosensory stimuli in the locust (*Homberg, 1994*). In the moth, recordings from neurons innervating the LAL have a characteristic 'flip-flop' firing property, which is thought to mediate walking commands (*Kanzaki et al., 1991b*; *Kanzaki et al., 1994*; *Mishima and Kanzaki, 1998*; *Mishima and Kanzaki, 1999*; *Wada and Kanzaki, 2005*). More recent work has suggested a functional organization whereby the neurons in the upper division of the LAL receive convergent input from the protocerebrum and neurons in the lower division generate locomotor command (*Namiki et al., 2014*; *Rayshubskiy et al., 2020*).

Our data show that the MB network converges with the protocerebrum input, thereby providing an opportunity for MBONs to indirectly influence descending motor outputs. We also demonstrate that two MBONs (γ3β′1 and γ2α′1) synapse on the same subset of LAL and FSB cells, revealing a convergent circuit that connects both structures. Further, in support of our anatomical observations, optogenetic activation of MBON γ2α′1 resulted in activation of both LAL and FSB layer four neurons. Given that MBON γ3β′1 is GABAergic, we did not perform the equivalent experiment for this neuron. Thus, understanding the functional consequences of these inhibitory connections will require further investigation. Interestingly, despite the fact that MBON γ3β′1 and γ2α′1 express different neurotransmitters and innervate different MB compartments, their manipulation has similar behavioral phenotypes: both promote sleep (*Aso et al., 2014b*; *Sitaraman et al., 2015a*; *Sitaraman et al., 2015b*), and artificial activation of either results in naive preference (*Aso et al., 2014b*). Further, activation of both MBON γ3β′1 and γ2α′1 together has an additive effect, which results in a significant increase in preference (*Aso et al., 2014b*).

The FSB and LAL have a well-established structural and functional connectivity. The LAL integrates information from the central complex, including the FSB, and provides a premotor signal to motor centers (*Wolff and Strausfeld, 2015b*). However, the behavioral significance of MBON γ3β′1 and γ2α′1 projections to both the FSB and LAL is less clear. Previous work demonstrated that activation of these MBONs while the flies explored an open arena did not significantly affect average speed or angular speed of individual flies (*Aso et al., 2014b*). By contrast, we found that inactivation of the putative downstream LAL neurons significantly increased overall activity of behaving flies in a social context and locomotor assay. Thus, the γ3β′1 and γ2α′1 MBONs may play a modulatory rather than required role in influencing behavioral response to an associated cue.

Recent work in *Drosophila* has demonstrated that the DANs that innervate MBON γ2α′1 regulate flight bout durations, and may provide a motivation signal via MBONs to the FSB and LAL to regulate motor activities (*Sharma and Hasan, 2020*). The LAL neurons receive multisensory input (*Namiki and Kanzaki, 2016*), and some LAL neurons make direct connections to descending neurons that control movement. Thus, this circuit organization enables integration of sensory signals with punishment or reward to direct the motion of the animal. In contrast, MBON connections with the FSB might play a role in providing context for flexible navigation, goal-directed actions, and memory-based navigation (*Le Möel and Wystrach, 2020*; *Yue and Mangan, 2020*).

If homology can be defined by shared expression of transcription factors and similar functional roles, the MB-FSB connection may be an appropriate model for understanding functional connections between the hippocampus and striatum (*Strausfeld and Hirth, 2013a*; *Wolff and Strausfeld, 2016*) and serve as an accessible model for understanding connectivity between more complex brain structures associated with memory. Further, given that the integrative relay role of the LAL is somewhat reminiscent of the vertebrate thalamus (*Strausfeld and Hirth, 2013a*), the complex connectivity between the MBONs, FSB, and LAL may also serve as an effective model for predicting and understanding functional connections between the hippocampus, striatum, and thalamus in the context of memory formation and action selection.

## Conclusions

Insects exhibit a great variety of complex behaviors, and significant effort has been devoted to understand the neural circuits that underlie these behaviors. The genetically accessible *Drosophila* is a great model for studying the interplay between circuit architecture and behavior owing to their complex yet tractable brains. The MB circuits and their role in learning and memory are among the most studied circuits in *Drosophila*. Although, the majority of these studies have focused on olfactory memory, it is clear that the MB plays a much broader role in insect behavior. In *Drosophila*, the MB is important for courtship memory (*McBride et al., 1999*; *Montague and Baker, 2016*; *Sitnik et al., 2003*), taste aversive memory (*Masek et al., 2015*) as well as visual memory (*Liu et al., 2006*; *Liu et al., 2016*; *Vogt et al., 2014*). In cockroaches, the MB has a role in place memory (*Mizunami et al., 1998*) and recent data in two different species of ants implicate the MB in spatial navigation to learned locations using visual cues (*Buehlmann et al., 2020*; *Kamhi et al., 2020*). In mammals, the hippocampus is similarly required for multiple forms of associative memory, including spatial navigation using visual cues (*Fenton et al., 2000a*; *Fenton et al., 2000b*; *Muller and Kubie, 1987*; *O'Keefe and Conway, 1978*; *Scaplen et al., 2014*). Thus, cross-species similarity in circuit organization and function may exist between the mushroom body and the hippocampus (*Wolff and Strausfeld, 2016*). However, such anatomical and functional cross-species comparisons can also be made between the mushroom body and the cerebellum (*Farris, 2011*; *Litwin-Kumar et al., 2017*; *Modi et al., 2020*), suggesting that similar convergent-divergent architecture may be a general principle of structures that encode and update memories.

In this context, the implementation of *trans*-Tango to study the MB has high potential in the era of EM reconstruction of the *Drosophila* brain. Through examination of the circuit connectivity in several individuals, easily afforded by *trans*-Tango, the value of the EM reconstruction data could be augmented by overlaying on it potential nuanced differences between individuals. In addition, *trans*-Tango-mediated discoveries in the fly could help illuminate principles of circuit organization in other species. Further, due to the modular design of *trans*-Tango, it could be readily reconfigured for other types of studies beyond circuit tracing. For example, only minimal modifications are required for implementing a configuration of *trans*-Tango for identifying the molecular composition of the postsynaptic partners. This strategy could be used to examine the evidence that MBONs stratify the FSB through different classes of peptidergic neurons (*Donlea et al., 2018*; *Kahsai et al., 2012*; *Kahsai and Winther, 2011*; *Nässel and Zandawala, 2019*; *Sareen et al., 2020*). Confirmation of these observations would suggest that the MB plays a critical role in regulating modulatory systems of a midbrain region that shares structural and functional commonalities with the vertebrate basal ganglia (*Strausfeld and Hirth, 2013a*; *Strausfeld and Hirth, 2013b*). Finally, through combining it with new genome editing strategies, *trans*-Tango could become a useful tool for comparative anatomy in other insects. This would enable the study of synaptic connections in non-model organisms and lead to deeper understanding of biological diversity (*Gantz and Akbari, 2018*).

Understanding how memories are formed, stored, and retrieved necessitates knowledge of the underlying neural circuits. Our characterization of the architecture of the neural circuits connecting the MB with downstream central complex structures lays the anatomical foundation for understanding the function of this circuitry. Our studies may also provide insight into general circuitry principles for how information is processed to form memories and update them in more complex brains.

# Materials and methods

## Key resources table

| Reagent type (species) or resource | Designation | Source or reference | Identifiers | Additional information |
|---|---|---|---|---|
| Genetic reagent (*D. melanogaster*) | *y[1]w[*]* | *Pfeiffer et al., 2008* | | |
| Genetic reagent (*D. melanogaster*) | *UAS-shibire*<sup>ts1</sup> | *Pfeiffer et al., 2012* | FLYB: FBst0066600; RRID:BDSC_66600 | |
| Genetic reagent (*D. melanogaster*) | *LexAop-GCaMP6s, UAS-Chrimson* | Allan Wong (Janelia Research Campus) | N/A | 13xLexAop2-Syn21-opGCaMP6s in su(Hw)attP8, 10xUAS-Syn21-Chrimson88-tdTomato-3.1 in attP18 |
| Genetic reagent (*D. melanogaster*) | *trans-Tango* | *Talay et al., 2017* | FLYB: FBst0077124; RRID:BDSC_ 77124 | *trans*-Tango in attP40 |
| Genetic reagent (*D. melanogaster*) | *UAS-myrGFP, QUAS-mtdTomato* | *Talay et al., 2017* | FLYB: FBst0077479; RRID:BDSC_77479 | 10xUAS-myrGFP, 5xQUAS-mtdTomato(3xHA) in su(Hw)attP8 |
| Genetic reagent (*D. melanogaster*) | *UAS-CD2, QUAS-mtdTomato* | This study | N/A | 10xUAS-CD2, 5xQUAS-mtdTomato(3xHA) in su(Hw)attP8 |
| Genetic reagent (*D. melanogaster*) | *brp-SNAP* | *Kohl et al., 2014* | FLYB: FBst0058397; RRID:BDSC_ 58397 | brp[SNAPf-tag]/Cyo |
| Genetic reagent (*D. melanogaster*) | *LexAop-GFP* | *Pfeiffer et al., 2010* | FLYB: FBst0032203; RRID:BDSC_32203 | 13XLexAop2-mCD8::GFP in attP2 |
| Genetic reagent (*D. melanogaster*) | *MB002B-split-GAL4* | *Aso et al., 2014a* | FlyLight Robot ID: 2135053 RRID:BDSC_68305 | MBON β'2mp (4), γ5β'2a (2) |
| Genetic reagent (*D. melanogaster*) | *MB011B-split-GAL4* | *Aso et al., 2014a* | FlyLight Robot ID: 2135062 RRID:BDSC_68294 | MBON γ5β'2a (4), β'2mp (3), β'2mp_bilateral (3) |
| Genetic reagent (*D. melanogaster*) | *MB018B-split-GAL4* | *Aso et al., 2014a* | FlyLight Robot ID: 2135069 RRID:BDSC_68296 | MBON α'2 (4) |
| Genetic reagent (*D. melanogaster*) | *MB026B-split-GAL4* | *Aso et al., 2014a* | FlyLight Robot ID: 2135077 RRID:BDSC_68300 | MBON α'1 (3), α'3ap (2) |
| Genetic reagent (*D. melanogaster*) | *MB027B-split-GAL4* | *Aso et al., 2014a* | FlyLight Robot ID: 2135078 RRID:BDSC_68301 | MBON α'3ap (5), α'3 m (5) |
| Genetic reagent (*D. melanogaster*) | *MB050B-split-GAL4* | *Aso et al., 2014a* | FlyLight Robot ID: 2135100 RRID:BDSC_68365 | MBON α'1 (2), α2sc (4) |
| Genetic reagent (*D. melanogaster*) | *MB051B-split-GAL4* | *Aso et al., 2014a* | FlyLight Robot ID: 2135101 RRID:BDSC_68275 | MBON α'2 (1), γ2α'1 (4) |
| Genetic reagent (*D. melanogaster*) | *MB051C-split-GAL4* | *Aso et al., 2014a* | FlyLight Robot ID: 2135136 RRID:BDSC_68249 | MBON α'2 (1), γ2α'1 (3) |
| Genetic reagent (*D. melanogaster*) | *MB057B-split-GAL4* | *Aso et al., 2014a* | FlyLight Robot ID: 2135106 RRID:BDSC_68277 | MBON β'1 (3) |
| Genetic reagent (*D. melanogaster*) | *MB074C-split-GAL4* | *Aso et al., 2014a* | FlyLight Robot ID: 2135122 RRID:BDSC_68282 | MBON β'2mp (4), β2β'2a (3), γ5β'2a (1) |

*Continued on next page*

*Continued*

| Reagent type (species) or resource | Designation | Source or reference | Identifiers | Additional information |
|---|---|---|---|---|
| Genetic reagent (*D. melanogaster*) | MB077B- split-GAL4 | *Aso et al., 2014a* | RRID:BDSC_68283 | MBON γ2α′1 (4) |
| Genetic reagent (*D. melanogaster*) | MB077C- split-GAL4 | *Aso et al., 2014a* | FlyLight Robot ID: 2135125 RRID:BDSC_68284 | MBON γ2α′1 (3) |
| Genetic reagent (*D. melanogaster*) | MB080C- split-GAL4 | *Aso et al., 2014a* | FlyLight Robot ID: 2135128 RRID:BDSC_68285 | MBON α2sc (2) |
| Genetic reagent (*D. melanogaster*) | MB082C- split-GAL4 | *Aso et al., 2014a* | FlyLight Robot ID: 2135130 RRID:BDSC_68286 | MBON α′2 (3), α3 (5) |
| Genetic reagent (*D. melanogaster*) | MB083C- split-GAL4 | *Aso et al., 2014a* | FlyLight Robot ID: 2135131 RRID:BDSC_68287 | MBON γ3 (5), γ3β′1 (5) |
| Genetic reagent (*D. melanogaster*) | MB093C- split-GAL4 | *Aso et al., 2014a* | FlyLight Robot ID: 2135141 RRID:BDSC_68289 | MBON α′2 (4) |
| Genetic reagent (*D. melanogaster*) | MB110C-split-GAL4 | *Aso et al., 2014a* | FlyLight Robot ID: 2135158 RRID:BDSC_68262 | MBON γ3 (5), γ3β′1 (5) |
| Genetic reagent (*D. melanogaster*) | MB210B-split-GAL4 | *Aso et al., 2014a* | FlyLight Robot ID: 2135258 RRID:BDSC_68272 | MBON γ5β′2a (1), β′2mp (4), β2β′2a (3) |
| Genetic reagent (*D. melanogaster*) | MB298B-split-GAL4 | *Aso et al., 2014a* | FlyLight Robot ID: 2135346 RRID:BDSC_68309 | MBON γ4>γ1γ2 (4) |
| Genetic reagent (*D. melanogaster*) | MB310C-split-GAL4 | *Aso et al., 2014a* | FlyLight Robot ID: 2135358 RRID:BDSC_68313 | MBON α1 (5) |
| Genetic reagent (*D. melanogaster*) | MB399B-split-GAL4 | *Aso et al., 2014a* | FlyLight Robot ID: 2501738 RRID:BDSC_68369 | MBON β2β′2a (2) |
| Genetic reagent (*D. melanogaster*) | MB433B-split-GAL4 | *Aso et al., 2014a* | FlyLight Robot ID: 2501774 RRID:BDSC_68324 | MBON β1>α (3), γ4>γ1γ2 (4) |
| Genetic reagent (*D. melanogaster*) | MB434B-split-GAL4 | *Aso et al., 2014a* | FlyLight Robot ID: 2501775 RRID:BDSC_68325 | MBON β1>α (4), γ4>γ1γ2 (4) |
| Genetic reagent (*D. melanogaster*) | MB542B-split-GAL4 | *Aso et al., 2014a* | FlyLight Robot ID: 2501887 RRID:BDSC_68372 | MBON α′1 (1), α′3 m (2), α2p3p (2) |
| Genetic reagent (*D. melanogaster*) | GMR47H09-LexA | *Pfeiffer et al., 2013* | FLY: FBtp0088666 RRID:BDSC_53482 | |
| Genetic reagent (*D. melanogaster*) | VT055139-LexA | *Tirian and Dickson, 2017* | N/A | |
| Genetic reagent (*D. melanogaster*) | VT018476-lexA | *Bidaye et al., 2014* | N/A | |
| Genetic reagent (*D. melanogaster*) | SS01308-split GAL4 | Janelia Research Campus | N/A | MBON γ5β′2a |
| Genetic reagent (*D. melanogaster*) | SS01143-split GAL4 | Janelia Research Campus | N/A | MBON β′2mp |
| Genetic reagent (*D. melanogaster*) | SS1194-split GAL4 | Janelia Research Campus | N/A | MBON α2sc |
| Genetic reagent (*D. melanogaster*) | SS32219-split GAL4 | Janelia Research Campus | N/A | Lateral Accessory Lobe |

*Continued on next page*

*Continued*

| Reagent type (species) or resource | Designation | Source or reference | Identifiers | Additional information |
|---|---|---|---|---|
| Genetic reagent (*D. melanogaster*) | *SS32230-split GAL4* | Janelia Research Campus | N/A | Lateral Accessory Lobe |
| Antibody | α-GFP (Rabbit polyclonal) | Life Tech | Cat #A11122 RRID:AB_221569 | (1:1000) |
| Antibody | α-HA (Rat monoclonal) | Roche | Cat #11867423001 RRID:AB_390918 | (1:100) |
| Antibody | α-GFP (Chicken polyclonal) | Clontech | Cat #ab13970 RRID:AB_300798 | (1:2000) |
| Antibody | α-DS (Rabbit monoclonal) | Clontech | Cat #632496 RRID:AB_10013483 | (1:1000) |
| Antibody | α-CD2 (Mouse monoclonal) | Bio-Rad | Cat #MCA154GA RRID:AB_566608 | (1:100) |
| Antibody | α-TH (Mouse monoclonal) | Immunostar | Cat #22941 RRID:AB_572268 | (1:500) |
| Antibody | Goat α-Mouse AF647 (polyclonal) | Thermo Fisher | Cat #A21235 RRID:AB_2535804 | (1:1000) |
| Antibody | Goat α-Rabbit AF488 (polyclonal) | Life Tech | Cat #A11034 RRID:AB_2576217 | (1:400) |
| Antibody | Goat α-Rat AF568 (polyclonal) | Life Tech | Cat #A11077 RRID:AB_2534121 | (1:400) |
| Antibody | Goat α-Chicken AF488 (polyclonal) | Life Tech | Cat #A11039 RRID:AB_2534096 | (1:400) |
| Antibody | Goat α-Rabbit AF568 (polyclonal) | Life Tech | Cat #A11011 RRID:AB_143157 | (1:400) |
| Software | Adobe Illustrator CC | Adobe | RRID:SCR_014199 | |
| Software | ZEN | Carl Zeiss Microscopy | Version 2.1 (blue edition) RRID:SCR_013672 | |
| Software | Fiji | http://fiji.sc | RRID:SCR_002285 | |

## Fly strains

All *Drosophila melanogaster* lines were raised at 18°C on standard cornmeal-agar media with tego-sept antifungal agent and in humidity-controlled chambers under 14/10 hr light/dark cycles. SS lines were previously made in the Rubin lab in collaboration with the Janelia FlyLight team and the Janelia Fly facility. For a list of fly lines used in the study, see the Key Resource Table.

## Generation of transgenic UAS-CD2, QUAS-mtdTomato lines

Gibson Assembly was used to generate the plasmid UAS-CD2_QUAS-mtdTomato(3xHA). The DNA sequence encoding Rattus norvegicus CD2 (NP_036962.1) was codon optimized for *Drosophila melanogaster* and synthesized by Thermo Fisher Scientific, USA. This sequence was subsequently amplified using primers 5'-atcctttacttcaggcggccgcggctcgagaatcaaaATGCGCTGCAAGTTCCTG-3' and 5'-agtaaggttccttcacaaagatcctctagaTTAGTTGGGTGGGGGCAG-3' to obtain the insert fragment. To generate the vector fragment, the *trans*-Tango reporter plasmid (UAS-myrGFP_QUAS-mtdTomato (3xHA)) (*Talay et al., 2017*) was digested with XhoI and XbaI. Insert and vector fragments were ligated using HiFi DNA Assembly Kit (New England Biolabs, USA) following manufacturer's instructions. The resultant plasmid was integrated at the su(Hw)attP8 site via PhiC31-mediated recombination.

### *trans*-Tango immunohistochemistry

Flies were dissected at 15–20 days post-eclosion using methods adapted from FlyLight Protocols (https://www.janelia.org/project-team/flylight/protocols). Flies were anesthetized with temperature, dewaxed in 70% ethanol, rinsed in Schneider's Insect Medium (S2) and dissected on a Sylgard pad with cold S2. Within 20 min of dissection, collected brains were transferred to 2% paraformaldehyde (PFA) in S2 and incubated for 55 min at room temperature. After fixation, brains were rinsed with phosphate buffered saline with 0.5% Triton X-100 (PBT) for 15 min at room temperature. Washes were repeated four times before storing the brains overnight in 0.5% PBT at 4°C. For chemical tagging in brp-SNAP+ brains, PBT was removed and SNAP substrate diluted in PBT (SNAP-Surface649, NEB S9159S; 1:1000) added. Brains were incubated for 1 hr at room temperature and rinsed with PBT (3 times for 10 min). Brains were then blocked in 5% GS (Goat Serum) diluted in PBT for 90 min at room temperature. Brains were then incubated in primary antibodies diluted in 5% GS/PBT for 4 hr at room temperate and then at 4°C for two overnights. After primary antibody incubation, brains were washed four times for 10 min with 0.5% PBT before incubating in secondary antibodies diluted in 5% GS/PBT at 4°C for two overnights. Samples were then rinsed and washed four times for 15 min in 0.5% PBT at room temperature and prepared for DPX mounting. Briefly, brains were fixed a second time in 4% PFA in PBS for 4 hr at room temperature and then washed four times in PBT for 15 min at room temperature. Brains were rinsed for 10 min in PBS, placed on PLL-dipped cover glass, and dehydrated in successive baths of ethanol for 10 min each. Brains were then soaked three times in xylene for 5 min each and mounted using DPX.

### Genetic overlap analysis

MBON split-GAL4 'C' lines which have the DNA-binding domain (in attP2) and activation domain (in VK00027) recombined on the 3rd chromosome were crossed to newly generated *trans*-Tango reporter flies where the 10xUAS-myrGFP was replaced with 10xUAS-CD2, and 13xLexAOp-mCD8::GFP was inserted into attP2. This enabled the visualization of the starter MBONs, the postsynaptic *trans*-Tango signal, and the LexA lines simultaneously.

### Microscopy and image analysis

Confocal images were obtained using a Zeiss, LSM800 (Brown University) and LSM710 (Janelia Research Campus) with ZEN software (Zeiss, version 2.1) with auto Z brightness correction to generate a homogeneous signal and were formatted using Fiji software (http://fiji.sc). Whole brains were scanned using a 40x objective in four overlapping tiles and then stitched together in the ZEN software.

TH+ cells, and cells with overlapping TH and *trans*-Tango signal were counted by blinded experimenter using the Cell Counter plugin in FIJI (https://imagej.net/Cell_Counter). We counted the total number of TH+ cells that co-localized with *trans*-Tango labeled cells in each hemibrain starting at the most anterior surface of the brain and continued to count TH+ cells until we reached the protocerebral anterior lateral (PAL) cluster which were identified by their cell body size. We did not identify any co-localized cells within or posterior to the PAL cluster.

Images were prepared for publication in FIJI and Adobe Illustrator with no external manipulation aside from cropping to demonstrate higher resolution. All figures were generated using Adobe Illustrator CC.

### Brain registration and tracing postsynaptic connections

Brains were registered as previously described (*Aso et al., 2014a*). Postsynaptic connections of registered brains were segmented in VVD Viewer (https://github.com/takashi310/VVD_Viewer; *Wan et al., 2009*; *Wan et al., 2012*) and saved as .nrrd files. Segmented files of postsynaptic signal for each MBON were multiplied by 34 binary masks of each central brain region in a custom written Matlab program to calculate the distribution of postsynaptic signal across brain regions. Signal within each brain was normalized by calculating a Z-score, or the number of standard deviations above or below the mean signal, for each brain regions. Heatmaps were generated in RStudio.

## Calcium imaging protocol and analysis

All functional imaging experiments were performed ex-vivo from brains of 1- to 4-day-old male or female brains on an Ultima two-photon laser scanning microscope (Bruker Nanosystems) equipped with galvanometers driving a Chameleon Ultra II Ti-Sapphire laser. Images were acquired with an Olympus 60x, 0.9 numerical aperture objective at $512 \times 512$ pixel resolution.

Flies were placed on food containing 400 µM all trans-retinal for 18–36 hr prior to dissection. Brains were dissected in saline (108 mM NaCl, 5 mM KCL, 2 mM CaCl2, 8.2 mM MgCl2, 4 mM NAHCO3, 1 mM NaH2PO4, 5 mM trehalose, 10 mM sucrose, 5 mM HEPES, pH 7.5 with osmolarity adjusted to 275 mOsm), briefly (45 s) treated with collagenase (Sigma #C0130) at 2 mg/mL in saline, washed, and then pinned with fine tungsten wires in a thin Sylgard sheet (World Precision Instruments) in a 35 mm petri dish (Falcon) filled with saline. MBONs were stimulated with 400-500ms of 627 nm LED. For recordings in the LAL (VT018476 and VT055139) ROI were positioned over SMP. For recordings in the FSB (476H09) ROIs were positioned over SMP or FSB.

All image processings were done using FIJI/ImageJ (NIH). Further analysis was performed using custom scripts in ImageJ, Microsoft Excel, and RStudio. Normalized time series of GCaMP fluorescence were aligned to the time point when the opto-stimulus was applied for each replicate.

## Behavioral experiments

Locomotor activity was evaluated in a 37 mm diameter circular open field area as described previously (*Scaplen et al., 2019*). Briefly, for thermogenetic inactivation, 10 flies were placed into arena chambers and placed in a 30°C incubator for 20 min prior to testing. The arena was then transferred to a preheated (30°C) light sealed box and connected to a humidified air delivery system. Flies were given an additional 15 min to acclimate to the box before recordings began. Group activity was recorded (33 frames/s) for 20 min. Recorded .avi files of fly activity were processed by FFMPEG and saved as .mp4. Individual flies were tracked using Caltech Flytracker (*Eyjolfsdottir et al., 2014*) to obtain output features such as position, velocity, and angular velocity. Feature based activity was averaged across within each genotype and plots were generated in RStudio.

## Acknowledgements

KMS and KRK were supported by the Smith Family Award Program for Excellence in Biomedical Research (KRK) and the NIH grant R01AA024434 (KRK). KMS and KRK were also supported by a grant from NIH (5P20GM103645) to the Carney Institute for Brain Science Center for Nervous System Function COBRE. GB, MT, AS and JDF were supported in part by NIH grants R01DC017146 and R01MH105368. We thank Vanessa Ruta for many fruitful discussions, advice on the manuscript and experiments, as well as support of RC. We thank Janelia Fly facility for help with fly husbandry, and the FlyLight Project team for help with brain dissections, histological preparations, and confocal imaging performed at Janelia Research campus. We thank the Rubin lab (Janelia Research Campus), Janelia FlyLight team, and the Janelia Fly facility for their assistance in making SS lines used in this paper. We also thank Hideo Otsuna and Takashi Kawase (Janelia Research Campus) for helpful software advice, Arif Hamid (Brown University) for helpful programming advice, and Gina Chieffallo for initial anatomical characterizations. This work was made possible in part by software funded by the NIH: Fluorender: An Imaging Tool for Visualization and Analysis of Confocal Data as Applied to Zebrafish Research, R01-GM098151-01.

## Additional information

### Funding

| Funder | Grant reference number | Author |
| --- | --- | --- |
| Richard and Susan Smith Family Foundation | | Kristin M Scaplen<br>Karla R Kaun |
| National Institute on Alcohol Abuse and Alcoholism | R01AA024434 | Karla R Kaun<br>Kristin M Scaplen |
| National Institute of General | P20GM103645 (8278) | Kristin M Scaplen |

| Medical Sciences | | Karla R Kaun |
| National Institute on Deafness and Other Communication Disorders | R01DC017146 | Mustafa Talay<br>John D Fisher<br>Altar Sorkaç<br>Gilad Barnea |
| National Institute of Mental Health | R01MH105368 | Mustafa Talay<br>John D Fisher<br>Altar Sorkaç<br>Gilad Barnea |

The funders had no role in study design, data collection and interpretation, or the decision to submit the work for publication.

### Author contributions

Kristin M Scaplen, Conceptualization, Data curation, Software, Formal analysis, Validation, Investigation, Visualization, Methodology, Writing - original draft, Project administration, Writing - review and editing; Mustafa Talay, Conceptualization, Resources, Data curation, Formal analysis, Validation, Investigation, Visualization, Methodology, Writing - review and editing; John D Fisher, Investigation, Visualization, Writing - review and editing; Raphael Cohn, Investigation, Writing - review and editing; Altar Sorkaç, Methodology, Writing - review and editing; Yoshi Aso, Resources, Validation, Writing - review and editing; Gilad Barnea, Conceptualization, Resources, Supervision, Funding acquisition, Methodology, Project administration, Writing - review and editing; Karla R Kaun, Conceptualization, Resources, Supervision, Funding acquisition, Writing - original draft, Project administration, Writing - review and editing

### Author ORCIDs

Kristin M Scaplen  https://orcid.org/0000-0001-7493-1420
Yoshi Aso  https://orcid.org/0000-0002-2939-1688
Gilad Barnea  https://orcid.org/0000-0001-6842-3454
Karla R Kaun  https://orcid.org/0000-0002-8756-9528

### Decision letter and Author response

Decision letter https://doi.org/10.7554/eLife.63379.sa1
Author response https://doi.org/10.7554/eLife.63379.sa2

## Additional files

### Supplementary files

- Supplementary file 1. Statistical analysis summary.
- Transparent reporting form

### Data availability

Source data files for Figure 1 are available on the Brown University Digital Repository https://doi.org/10.26300/mttr-r782.

The following dataset was generated:

| Author(s) | Year | Dataset title | Dataset URL | Database and Identifier |
| --- | --- | --- | --- | --- |
| Scaplen KM, Talay M, Fisher JD, Cohn R, Sorkaç A, Aso Y, Barnea G, Kaun KR | 2021 | Data from "Transsynaptic mapping of *Drosophila* mushroom body output neurons" | https://doi.org/10.26300/mttr-r782 | Brown Digital Repository, 10.26300/mttr-r782 |

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
