## [Decision Letter]

**Acceptance summary:**

In this manuscript, the authors describe the postsynaptic targets of MBONs using the relatively new trans-Tango system, including identification of dopaminergic cells and divergent or convergent projections to several forebrain structures. They further confirm the functional connectivity of one MBON to downstream targets. This study will be of broad interest because mapping downstream targets of the MBONs will impact models of learning, memory, and behavior output.

**Decision letter after peer review:**

Thank you for submitting your article "Transsynaptic mapping of *Drosophila* mushroom body output neurons" for consideration by *eLife*. Your article has been reviewed by two peer reviewers, and the evaluation has been overseen by a Reviewing Editor and K VijayRaghavan as the Senior Editor. The following individual involved in the review of your submission has agreed to reveal their identity: Nicholas Strausfeld (Reviewer #1).

The reviewers have discussed the reviews with one another and the Reviewing Editor has drafted this decision to help you prepare a revised submission.

Usually, when we ask for revisions before resubmission we have a consolidated review. In this case, both the reviews are separately appended. As you can see, we are positive about the work itself, but we have concerns about its effective communication. Please address all the concerns of both reviewers and submit a revised manuscript.

*Reviewer #1:*

This paper is revolutionary, which is why my review far exceeds 500 words.

In the not so distant past, the pipe-dream of any card-carrying neurologist/neuroanatomist was a method that would resolve a functional pathway expressed by its constituent morphologically identifiable and synaptically contiguous nerve cells. For those working on arthropod central nervous systems, the closest to achieving that was by using cobalt ions that move from one neuron to another across mixed chemical/gap-junction synapses to reveal physiologically connected neurons, such as those comprising the dipteran giant fiber system. Connections of lobula plate wide-field directional-selective neurons to a subset of identified descending neurons supplying the neck motor – another ultrafast electrical pathway – was likewise resolved.

And that was basically the end of it, until now. The invention of trans-Tango makes a distant fata morgana a reality. It fulfils historical aspirations of being able to realize rules governing how a brain works as an integrative whole. The potential of this really fantastic method, demonstrated in its original Neuron paper, was its fidelity in resolving in "one go" a well-established circuit, the structure of which had been obtained by time-consuming conventional means. For anyone working on comparative neural systems, the Neuron paper was a revelation: not simply because of the promise of what it demonstrated but, more prosaically, also because of its descriptive clarity and sensible use of summary figures. The authors of that Neuron paper, some of which are also on the present manuscript, appeared to have been cognizant of a potentially broad audience, one that includes students of brain anatomy and evolution in the broadest sense.

Thus, when I received the present manuscript to review, I was thrilled to see that trans-Tango had now been applied to the most researched part of the panarthropod brain, namely the mushroom bodies and their associated neuropils. However, to my dismay, in contrast to the original Neuron paper, which, as remarked, spoke to a broad audience, the way in which data are presented in this manuscript seems to exclusively for the consumption by a highly restricted coterie dedicated to elucidating the fruitfly brain. Now, those authors might protest my objecting to this on the grounds that it's immaterial to them whether those working outside *Drosophila* read this work. But it is not immaterial to the outside; and the reason that it's not is because this manuscript is revolutionary. It's a neuroanatomical tour de force like no other, and as such must be readily accessible (and intelligible) to those who are not particularly *Drosophila*-centric. The problem I have with this account is not its science. The findings are unimpeachable, gorgeous. It is with a manuscript that requires translation to scientific vernacular.

I'm sorry to say that the way this work is presented will challenge and may well discourage being read by anyone working on other types of brain; be it arthropod, fish, or whatever. I do not work on *Drosophila*, but because my game is comparative and evolutionary neuroanatomy, I can fully appreciate the present analysis and its findings and can also see how this work can directly impact a larger body of research on arthropod brains and behavior. The enlarged images enable someone working on another insect brain to recognize corresponding neurons and pathways neurons depicted. Scientists studying other brains will likely have well-formed expectations about where their neurons project, and to what subsequent neurons they possibly contact. Definitive data from the fruitfly could thus provide serious support for work on other species. The present authors can't count on a readership that is as informed about the fly's brain as they are, so they need to help their non-Drosophilist colleagues understand the principal arrangements of MBONs, DANs et al. and describe in plain language how functional types are segregated into cohorts and how these underpin the functional logic of downstream pathways.

Aso and colleagues have already provided us with the functional logic of immediate mushroom body connections in previous *eLife* publications and their summary figures there serve as excellent guides even for researchers not directly involved in *Drosophila* neurobiology (even researchers working on the brains of non-insect mandibulates!). Here, in the present work, even though Figures 4 and 7 provide three schematics, for the non-specialist a comprehensive summary figure would be tremendously helpful to recognize putative connections that might be recognized in other brains. For example, a pivotal finding of this manuscript is that a subset of mushroom body efferents terminates in discrete domains of the superior protocerebrum where they interact with systems of local interneurons. These are shown as participating in connections involving the MBON terminals and dendritic fields of axonal neurons that define stratified synaptic levels of the fan-shaped body. I was delighted by this beautiful demonstration because it confirms earlier work from my laboratory resolving in the blowfly the same organizational relationships; albeit not by transynaptic technology but intracellular recordings and cobalt fills on a blowfly (DOI 10.1002/cne.23094). Others studying the organization of the bee brain, or the brains of crickets, butterflies, beetles, etc., would welcome similar corroborations and they would likely get this with reference to the present paper and future trans-Tango resolutions of *Drosophila* circuitry. This projection is a compelling reason for the present manuscript's descriptions to be widely accessible to other researchers who are not Drosophlists. But where, in the present work, is the common language and comprehensive schematics?

My opinion boils down to this: the data are exquisite but they risk being undermined by the vehicle that conveys them. A description lacking in appeal to workers in other fields will strangle their attention. I accept that the body of the results has to be written in Technodrosophalese. But why the Discussion too? After its opening generalities, as it now stands the Discussion addresses just a small subgroup of *Drosophila* cognoscenti. Its narrative may be undecipherable for anybody expecting some overarching conclusions or interpretations that have broader import. One can't expect someone analyzing an ant brain to be familiar with the density of acronyms here, each signifying a different neuron's identity. This style of presentation restricts cross-fertilization with studies of brains other than those of *Drosophila*. For example, questions about memory acquisition and its behavioral manifestation in an ant are likely to be more challenging than anything envisaged for the fly. This is suggested by the very recent confirmation by two independent groups (Current Biol. October 2020) that the MB vertical lobes are required for an ant's spatial memory during visual exploration (as distinct from path integration) and the memory of object location. Those experiments imply that visual space and the valences of its visual features are encoded in a manner that preserves the memory of their spatial relationships. Is this possible within the MB cellular matrix? What features of the MB outputs would allow such a representational map? These questions get us close to considerations about the hippocampus and its possible analogues/homologues; the questions trigger, or at least it should trigger, serious thought about common principles that occur across not just species but phyla too. And while everyone will understand that detailed technical descriptions are part and parcel of the present paper's neurogenetic sophistication, that there is little relief from technospeak in the Discussion dampens the allure of an otherwise splendid work. Something needs to be done to make all of this broadly attractive and thus palatable to more than a highly specialized audience. Add to these challenges another deterrent: that the illustrations – the manuscript's essential data – deserve more than postage-size reproduction (yes, one can enlarge the images; but their manuscript dimensions are really off-putting). I hope that *eLife* will allow the selected image to be shown as "full size" supplements.

So, to repeat ad nauseam: a potentially broad readership needs a digestible narrative. It doesn't have to be wholly in the vernacular but needs to be clear. Its citations properly deployed. I have the impression that the authors chuck in a few often-deployed citations as cosmetic elements (one close to home: Wolff and Strausfeld, 2015 is a comparative study of DC0-positive MBs across arthropods, not a study of the LAL). The Discussion might benefit the reader if it sensibly suggests how trans-Tango-derived discoveries in the fly might illuminate questions arising from studies of other species. An example: splendid studies from the Nässel lab in Stockholm demonstrate FB stratification imposed by different classes of peptidergic neurons whose dendrites arborize in the SMP, SIP, and SLP. The present trans-Tango manuscript now confirms that these superior protocerebral neuropils receive inputs from MBONs and demonstrates the synaptic contiguity of MNBONs and FB afferents. This suggests that the MBs must play a critical role in regulating modulatory systems in a midbrain center comparable to the vertebrate basal ganglia. Mentioning this kind of bridge to other studies would to bring them into the fold. Yet such reflections are missing in the Discussion.

The Introduction is another difficulty, which I can approach line-by-line. It may seem frightfully school-masterish but it's meant to resolve ambiguities and some clumsiness. It's often forgotten that a lucid Introduction is a crucial element of a paper (as nicely demonstrated by the trans-Tango Neuron paper). An Introduction should seduce the reader to embrace the body of the paper. The present Introduction is chilling.

"The brain comprises intricate neural networks in which information iteratively converges and diverges to support learning, memory, and behavioral flexibility. Knowledge of the neural connectivity that underlies these networks is essential to our understanding of how the brain functions." My comment may seem pedantic, but this sentence not only waffles but suggests that such networks are already known. It might cause a reader to wonder why they would require further investigation. I'm sure this wasn't the authors' intent and would suggest a simpler and more direct opening with appropriate citations and a declaration about what of importance is still up in the air.

"iteratively." Is this being used in its computational sense: arrangements that recycle information to achieve greater accuracy in obtaining a less noisy outcome?

"…and has a well-established role in olfactory learning and memory…" True, but because we know that it supports much more than this, it is time that those other properties are given as much prominence particularly after the already mentioned (by the authors) multimodal nature of the MBs.

"…parallel axonal fibers…" K-cells processes are neither parallel nor axonal. They extend together within their parent fascicle; but please look at published descriptions of those. In all the lobes except the core, the K-cell processes crisscross and tangle; they send off collaterals to providing local systems of intrinsic Hebbian-like circuitry. And, just because the outermost branches of Kenyon cells are predominantly postsynaptic receive sensory relays this doesn't make their extended processes axonal. K-cell processes in the lobes that provide elaborate local pre- and postsynaptic connections, even serial connections between inputs to the lobe and outputs from it. These arrangements within the lobes reveal K-cells to be functional local interneurons. Many insect lineages have K-cells that entirely lack "dendrites" (this "naked" organization is the ancestral condition), as is the case in mandibulates that are sister to Hexapoda.

"Kenyon cells that are intrinsic to the MB…." Kenyon cells by definition are MB intrinsic neurons. They don't exist external to the MB.

"They also receive organized valence-related input from dopamine neurons (DANs) and converge onto 34 different MB output neurons". This reads as if all the K-cells converge onto the same set of MBONs. But different MBON dendritic fields occupy different and specific domains down the extent of a lobe thereby demonstrating that subsets of K-cells must interact with different combinations of MBONs. This segregation to computational domains of inputs->local circuits->outputs is a cardinal feature of MBs, recognized for a very long time, beginning with Mobbs's work on the honey bee MB followed up by intracellular and neuroanatomical work on the cockroach MB. It's important for a broad understanding of the MBs to get this across-species (more accurately: across Pancrustacea) aspect of organization right, along with relevant citations.

"This architecture positions the MB as a high-level integration center for the representations of olfactory cues and their perceived valence." 'Positions' is a curious way to say that properties define the MBs as…etc… Yes, the MB is an integration center. But one needs to bear in mind that the *Drosophila* MB is the result of some serious evolved reduction of its ancestral multisensory character, as compared with MBs in related but more basal Schizophora, not to mention other insect lineages groups where MB has switched modalities (exclusively visual MB calyces in diving beetles). At this point in the Introduction papers might be cited that refer to other modality types: thus, "representations of sensory cues" seem a better descriptor than "representation of olfactory cues…".

"With the development of split-Gal4 lines that provide selective genetic access to precise neuronal populations (Aso et al., 2014a), detailed patterns of MB neural circuits have emerged over the past decade describing a compartmentalization of the MB lobes by arborization patterns of innervating dopamine neurons (DANs) and MBONs". I realize that people working on *Drosophila* show less attention to studies on other species, but if citations refer to the first recognition of such compartmentalization shouldn't this be mentioned in actual words?

"and several neuropil structures were identified as sites of convergence for the MBONs, including the." The reader will be confused by the authors' reference here to "convergence" because this sentence is summarizing divergence from the MB to various separate and distinct target neuropils.

"Within these convergent neuropil structures, MBON axons were proposed to synapse onto axons of other MBONs, and dendrites of DANs, interneurons and projection neurons. These convergent neuropil structures, however, are characterized by highly complex arborizations of dendrites and axons making it challenging to identify the specific neural components that receive synaptic input from various MBONs." I assume that a "convergent neuropil structure" is a more redundant way of saying neuropil. In any event, this passage needs a citation regarding MBONs synapsing onto other MBONs. If MBONs are proposed to synapse onto DANs, isn't this implying a feedback circuit to the MBs? It would be worth a mention, and I think there is evidence from intracellular work on the honeybee mushroom body. And what exactly is meant here by an "interneuron." A local anaxonal neuron? What is meant by a projection neuron? It's the accepted term for first-order relay neurons from an antennal lobe. Do the authors mean an axonal neuron? All these terms have precise definitions and are best used correspondingly.

What is the *Drosophila* field? An agricultural entity? A community? A community of people working on the fly brain?

"Although EM data offers synaptic resolution, it is labor-intensive and does not account for potential variability in synaptic connectivity that exists across animals." Don't the authors mean synaptic structural resolution? Aren't synaptic strengths, sign, modulation, still matters for speculative interpretation and not usually evidential from EM images?

"We found that MB efferent pathways are highly interconnected, including several points of convergence among MBONs". The reader won't know what this means: interconnected = cross-talk? Serial synaptic connections? Distributed? And what is a "point of convergence?" A neuropil? A synapse? The sentence's meaning won't obvious to the reader.

This in the Results, but needs a comment. "Nearly all MBONs have divergent connections across the dorsal brain regions "CRE, SMP, SIP, SLP, LH, as well as FSB, and LAL." I note that the Introduction the proper names are given. But the citation has to refer to the Ito et al. brain name paper.

Now I am finished.

*Reviewer #2:*

Thus far, mushroom body circuitry has been highly characterized, from inputs to outputs (mushroom body output neurons or MBONs). In this manuscript, the authors described the postsynaptic targets of MBONs using the relatively new trans-Tango system, including the identification of dopaminergic cells and divergent or convergent projections to several forebrain structures. They further confirmed the functional connectivity of one MBON to downstream targets. This study will be of broad interest because mapping downstream targets of the MBONs will impact models of learning, memory, and behavior output. While it is a sound study and an excellent companion to the current connectomic datasets, the impact is lessened by the descriptive nature and limited functional data.

1) Previous research has shown that MBONs target the same forebrain areas described here, including the fan-shaped body and lateral accessory lobe (Aso et al., 2014). My main concern is that the conclusions of this manuscript convey that connectivity between the MBs and protocerebral regions was not previously known.

2) Variability is demonstrated for cells postsynaptic to one cell. It would be useful to know to what extent inter-individual variability is observed in the cells that project to forebrain areas other than the FSB and LAL. Along those lines, was there any sexual dimorphism? Males and females were used from some lines, but why were only males used from other lines?

3) How was postsynaptic signal normalized within brains (Figure 2)? I could not find this in the Materials and methods.

4) The behavioral experiments described in Figure 7 seem disjointed from the rest of the study. Why inactivate the LAL and not the MBONs that project there as well? Also, please show the expression pattern for SS32230.

5) Although MBON g3b′1 is GABA-ergic, wouldn't be activating this neuron potentially depress activity in LAL and FSB neurons? The explanation of why this experiment was not done remains unclear.

---

## [Author Response]

Thank you for submitting your article "Transsynaptic mapping of *Drosophila* mushroom body output neurons" for consideration by eLife. Your article has been reviewed by two peer reviewers, and the evaluation has been overseen by a Reviewing Editor and K VijayRaghavan as the Senior Editor. The following individual involved in the review of your submission has agreed to reveal their identity: Nicholas Strausfeld (Reviewer #1).The reviewers have discussed the reviews with one another and the Reviewing Editor has drafted this decision to help you prepare a revised submission.We would like to draw your attention to changes in our revision policy that we have made in response to COVID-19 (https://elifesciences.org/articles/57162). Specifically, we are asking editors to accept without delay manuscripts, like yours, that they judge can stand as eLife papers without additional data, even if they feel that they would make the manuscript stronger. Thus the revisions requested below only address clarity and presentation.Usually, when we ask for revisions before resubmission we have a consolidated review. In this case, both the reviews are separately appended. As you can see, we are positive about the work itself, but we have concerns about its effective communication. Please address all the concerns of both reviewers and submit a revised manuscript.Reviewer #1:

We thank you for your thorough analysis of the prose of the manuscript and passion for making neuroanatomy accessible to a broad audience. We thoroughly appreciate your expertise in a broad range of central nervous systems and are grateful of the time you took to help improve our manuscript. As we certainly want our manuscript to be accessible to scientists who don’t study circuits in *Drosophila,* we have added summary schematics to Figures 3, 5, and 6 and added several comprehensive schematics which now comprise Figure 8. We have also significantly modified the Introduction and Discussion so it is accessible to a broader readership while still balancing the more nuanced comparisons with the EM dataset.

My opinion boils down to this: the data are exquisite but they risk being undermined by the vehicle that conveys them. A description lacking in appeal to workers in other fields will strangle their attention. I accept that the body of the results has to be written in Technodrosophalese. But why the Discussion too? After its opening generalities, as it now stands the Discussion addresses just a small subgroup of *Drosophila* cognoscenti. Its narrative may be undecipherable for anybody expecting some overarching conclusions or interpretations that have broader import. One can't expect someone analyzing an ant brain to be familiar with the density of acronyms here, each signifying a different neuron's identity. This style of presentation restricts cross-fertilization with studies of brains other than those of *Drosophila*. For example, questions about memory acquisition and its behavioral manifestation in an ant are likely to be more challenging than anything envisaged for the fly. This is suggested by the very recent confirmation by two independent groups (Current Biol. October 2020) that the MB vertical lobes are required for an ant's spatial memory during visual exploration (as distinct from path integration) and the memory of object location. Those experiments imply that visual space and the valences of its visual features are encoded in a manner that preserves the memory of their spatial relationships. Is this possible within the MB cellular matrix? What features of the MB outputs would allow such a representational map? These questions get us close to considerations about the hippocampus and its possible analogues/homologues; the questions trigger, or at least it should trigger, serious thought about common principles that occur across not just species but phyla too. And while everyone will understand that detailed technical descriptions are part and parcel of the present paper's neurogenetic sophistication, that there is little relief from technospeak in the Discussion dampens the allure of an otherwise splendid work. Something needs to be done to make all of this broadly attractive and thus palatable to more than a highly specialized audience. Add to these challenges another deterrent: that the illustrations – the manuscript's essential data – deserve more than postage-size reproduction (yes, one can enlarge the images; but their manuscript dimensions are really off-putting). I hope that eLife will allow the selected image to be shown as "full size" supplements.

We agree the images, particularly those in Figure 1 were too small. We have increased the size of the image and rotated the figure from portrait to landscape to accommodate the increase in size. As the reviewer suggests we have added full size images of each brain (Figure 1—figure supplement 2-23) and made raw.czi files of all brains available to the readership (https://doi.org/10.26300/mttr-r782). We have also rearranged most of the other figures in order to expand the images.

We appreciate the comments about the Introduction and Discussion sections aimed at making these sections more accessible to a broad readership. We thus have made substantial changes to the text. This includes removing the acronyms of neurons, significantly revising the MBON, FSB and LAL sections to incorporate more comparisons with other species and speculate about the function of the MB connectivity we reveal in the context of structures across species that encode memories.

So, to repeat ad nauseam: a potentially broad readership needs a digestible narrative. It doesn't have to be wholly in the vernacular but needs to be clear. Its citations properly deployed. I have the impression that the authors chuck in a few often-deployed citations as cosmetic elements (one close to home: Wolff and Strausfeld, 2015 is a comparative study of DC0-positive MBs across arthropods, not a study of the LAL). The Discussion might benefit the reader if it sensibly suggests how trans-Tango-derived discoveries in the fly might illuminate questions arising from studies of other species. An example: splendid studies from the Nässel lab in Stockholm demonstrate FB stratification imposed by different classes of peptidergic neurons whose dendrites arborize in the SMP, SIP, and SLP. The present trans-Tango manuscript now confirms that these superior protocerebral neuropils receive inputs from MBONs and demonstrates the synaptic contiguity of MNBONs and FB afferents. This suggests that the MBs must play a critical role in regulating modulatory systems in a midbrain center comparable to the vertebrate basal ganglia. Mentioning this kind of bridge to other studies would to bring them into the fold. Yet such reflections are missing in the Discussion.

We have added a paragraph in the conclusion section of the discussion that touches on how *trans*-Tango may be used to illuminate structural comparisons across insect species, to reveal the molecular identity of post-synaptic neurons, and to look at inter-individual variability and clarified the utility of *trans-*Tango elsewhere.

The Introduction is another difficulty, which I can approach line-by-line. It may seem frightfully school-masterish but it's meant to resolve ambiguities and some clumsiness. It's often forgotten that a lucid Introduction is a crucial element of a paper (as nicely demonstrated by the trans-Tango Neuron paper). An Introduction should seduce the reader to embrace the body of the paper. The present Introduction is chilling."The brain comprises intricate neural networks in which information iteratively converges and diverges to support learning, memory, and behavioral flexibility. Knowledge of the neural connectivity that underlies these networks is essential to our understanding of how the brain functions." My comment may seem pedantic, but this sentence not only waffles but suggests that such networks are already known. It might cause a reader to wonder why they would require further investigation. I'm sure this wasn't the authors' intent and would suggest a simpler and more direct opening with appropriate citations and a declaration about what of importance is still up in the air.

We appreciate your concern regarding the Introduction and have edited the first paragraph so it is clear and significantly edited other sections for clarity and readability as previously outlined.

"iteratively." Is this being used in its computational sense: arrangements that recycle information to achieve greater accuracy in obtaining a less noisy outcome? "…and has a well-established role in olfactory learning and memory…" True, but because we know that it supports much more than this, it is time that those other properties are given as much prominence particularly after the already mentioned (by the authors) multimodal nature of the MBs.

In simplifying this sentence, we have removed the word iteratively from the introductory sentence and broadened the description of the role of the MB in learning and memory to highlight its multimodal nature.

"…parallel axonal fibers…" K-cells processes are neither parallel nor axonal. They extend together within their parent fascicle; but please look at published descriptions of those. In all the lobes except the core, the K-cell processes crisscross and tangle; they send off collaterals to providing local systems of intrinsic Hebbian-like circuitry. And, just because the outermost branches of Kenyon cells are predominantly postsynaptic receive sensory relays this doesn't make their extended processes axonal. K-cell processes in the lobes that provide elaborate local pre- and postsynaptic connections, even serial connections between inputs to the lobe and outputs from it. These arrangements within the lobes reveal K-cells to be functional local interneurons. Many insect lineages have K-cells that entirely lack "dendrites" (this "naked" organization is the ancestral condition), as is the case in mandibulates that are sister to Hexapoda.

Parallel and axonal have been removed from the sentence to account for a more accurate description of the KC’s branching.

"Kenyon cells that are intrinsic to the MB…." Kenyon cells by definition are MB intrinsic neurons. They don't exist external to the MB.

We have updated this sentence accordingly.

"They also receive organized valence-related input from dopamine neurons (DANs) and converge onto 34 different MB output neurons". This reads as if all the K-cells converge onto the same set of MBONs. But different MBON dendritic fields occupy different and specific domains down the extent of a lobe thereby demonstrating that subsets of K-cells must interact with different combinations of MBONs. This segregation to computational domains of inputs->local circuits->outputs is a cardinal feature of MBs, recognized for a very long time, beginning with Mobbs's work on the honey bee MB followed up by intracellular and neuroanatomical work on the cockroach MB. It's important for a broad understanding of the MBs to get this across-species (more accurately: across Pancrustacea) aspect of organization right, along with relevant citations.

We have clarified this text and added citations to highlight a broad understanding the MB organization across species.

"This architecture positions the MB as a high-level integration center for the representations of olfactory cues and their perceived valence." 'Positions' is a curious way to say that properties define the MBs as…etc… Yes, the MB is an integration center. But one needs to bear in mind that the *Drosophila* MB is the result of some serious evolved reduction of its ancestral multisensory character, as compared with MBs in related but more basal Schizophora, not to mention other insect lineages groups where MB has switched modalities (exclusively visual MB calyces in diving beetles). At this point in the Introduction papers might be cited that refer to other modality types: thus, "representations of sensory cues" seem a better descriptor than "representation of olfactory cues…".

We have updated this sentence accordingly.

"With the development of split-Gal4 lines that provide selective genetic access to precise neuronal populations (Aso et al., 2014a), detailed patterns of MB neural circuits have emerged over the past decade describing a compartmentalization of the MB lobes by arborization patterns of innervating dopamine neurons (DANs) and MBONs". I realize that people working on *Drosophila* show less attention to studies on other species, but if citations refer to the first recognition of such compartmentalization shouldn't this be mentioned in actual words?

Thank you for pointing out the failure on our part in acknowledging the work of others outside of those working on *Drosophila*. We agree that this is a critical part of the story and regret this oversight. We have updated this section accordingly. We have significantly updated the Discussion to account for studies performed on other insects.

"and several neuropil structures were identified as sites of convergence for the MBONs, including the." The reader will be confused by the authors' reference here to "convergence" because this sentence is summarizing divergence from the MB to various separate and distinct target neuropils.

We recognize how one could view projections from the MB to LH, CRE, SMP, SIP and SLP as divergence from the MB (a single entity). It’s also true you can also describe divergence in terms of a single MBON projecting to separate distinct neuropil structures. However, here we are describing convergence from the perspective of the neuropil structures in that these regions receive input from multiple MBONs. We have clarified in the text and ask to keep this language to be consistent with previous descriptions (Aso et al. 2014).

"Within these convergent neuropil structures, MBON axons were proposed to synapse onto axons of other MBONs, and dendrites of DANs, interneurons and projection neurons. These convergent neuropil structures, however, are characterized by highly complex arborizations of dendrites and axons making it challenging to identify the specific neural components that receive synaptic input from various MBONs." I assume that a "convergent neuropil structure" is a more redundant way of saying neuropil. In any event, this passage needs a citation regarding MBONs synapsing onto other MBONs. If MBONs are proposed to synapse onto DANs, isn't this implying a feedback circuit to the MBs? It would be worth a mention, and I think there is evidence from intracellular work on the honeybee mushroom body. And what exactly is meant here by an "interneuron." A local anaxonal neuron? What is meant by a projection neuron? It's the accepted term for first-order relay neurons from an antennal lobe. Do the authors mean an axonal neuron? All these terms have precise definitions and are best used correspondingly.

Our intention was to draw attention to areas of neuropil that received inputs from multiple neuron types. However, the language we used was, as the reviewer pointed out, confusing. We have substantially revised this section, added references and defined our use of “interneuron”.

What is the *Drosophila* field? An agricultural entity? A community? A community of people working on the fly brain?

*Drosophila* field was used to describe the field of research which focuses on the fly brain. We have removed “*Drosophila*” from this sentence and talk about the “field” more broadly to avoid any apparent confusion.

"Although EM data offers synaptic resolution, it is labor-intensive and does not account for potential variability in synaptic connectivity that exists across animals." Don't the authors mean synaptic structural resolution? Aren't synaptic strengths, sign, modulation, still matters for speculative interpretation and not usually evidential from EM images?

Thank you for this observation. We did indeed mean synaptic structural resolution and have updated the sentence accordingly. We have also included the fact that synaptic strengths are not accounted for within EM analyses.

"We found that MB efferent pathways are highly interconnected, including several points of convergence among MBONs". The reader won't know what this means: interconnected = cross-talk? Serial synaptic connections? Distributed? And what is a "point of convergence?" A neuropil? A synapse? The sentence's meaning won't obvious to the reader.

We have updated this sentence accordingly.

This in the Results, but needs a comment. "Nearly all MBONs have divergent connections across the dorsal brain regions "CRE, SMP, SIP, SLP, LH, as well as FSB, and LAL." I note that the Introduction the proper names are given. But the citation has to refer to the Ito et al. brain name paper.

We have added this citation to the Introduction.

Now I am finished.

Thank you again for your extensive reading and analysis. We appreciate your contribution and hope to one day be as eloquent in our own prose when describing cross-species anatomical and functional frameworks.

Reviewer #2:Thus far, mushroom body circuitry has been highly characterized, from inputs to outputs (mushroom body output neurons or MBONs). In this manuscript, the authors described the postsynaptic targets of MBONs using the relatively new trans-Tango system, including the identification of dopaminergic cells and divergent or convergent projections to several forebrain structures. They further confirmed the functional connectivity of one MBON to downstream targets. This study will be of broad interest because mapping downstream targets of the MBONs will impact models of learning, memory, and behavior output. While it is a sound study and an excellent companion to the current connectomic datasets, the impact is lessened by the descriptive nature and limited functional data.1) Previous research has shown that MBONs target the same forebrain areas described here, including the fan-shaped body and lateral accessory lobe (Aso et al., 2014). My main concern is that the conclusions of this manuscript convey that connectivity between the MBs and protocerebral regions was not previously known.

Thank you for pointing this out. We have updated the Introduction and Discussion of results throughout the paper as a confirmation of previous predictions and an extension of the level of detail in describing connections.

2) Variability is demonstrated for cells postsynaptic to one cell. It would be useful to know to what extent inter-individual variability is observed in the cells that project to forebrain areas other than the FSB and LAL. Along those lines, was there any sexual dimorphism? Males and females were used from some lines, but why were only males used from other lines?

We would have loved to do a proper analysis of the inter-individual variability and even an analysis of the hemispheric variability for the postsynaptic connections within an animal. However, this would have required us to register and segment the signal from all of the brains we dissected and was beyond the scope of our study. Because analysis of FSB, LAL, and MBONs did not require brain registration and segmentation we restricted variability analyses to these structures.

With regard to why males and females were used, we collected data mainly from male flies as the output signal of *trans-*Tango is higher in males, given that the construct containing the reporter genes is incorporated at the su(Hw)attP8 site on the X chromosome. However, female brains were sometimes analyzed if the signal in males was difficult to discern. Further, females were used in some cases for more efficient registration on the fly template, as the commonly used template for brain registration is female. Unfortunately, we did not do a systematic comparison for sexual dimorphism as this was beyond the scope of our study.

We have updated the text to be more specific about whether the pattern was present in both sexes, when our analysis permitted. We have also made example raw.czi files of male and female brains for each line included in Figure 1 available to the readership (https://doi.org/10.26300/mttr-r782).

3) How was postsynaptic signal normalized within brains (Figure 2)? I could not find this in the Materials and methods.

We normalized the postsynaptic signal by calculating Z-Scores for each brain region. To clarify, we have added the following text to the Materials and methods “Signal within each brain was normalized by calculating a Z-score, or the number of standard deviations above or below the mean signal, for each brain regions”.

4) The behavioral experiments described in Figure 7 seem disjointed from the rest of the study. Why inactivate the LAL and not the MBONs that project there as well? Also, please show the expression pattern for SS32230.

We chose to prioritize LAL inactivation instead of MBONs g3b`1 or y2a`1 because previous work from Aso et al. 2014 showed that activation of these lines did not increase the mean speed (mm/s) or the mean angular speed (degrees/s). We have clarified the rationale of these experiments in the text and added an expression pattern for SS32230 to Figure 7.

5) Although MBON g3b′1 is GABA-ergic, wouldn't be activating this neuron potentially depress activity in LAL and FSB neurons? The explanation of why this experiment was not done remains unclear.

We thank the reviewer for their comment and interest in this circuit. Unfortunately, activating the GABAergic MBON g3b`1 and recording suppression in LAL and FSB isn’t as straightforward as recording suppression in LAL and FSB. This experiment assumes that there is a high level of basal activity in downstream structures such that when we activate the GABAergic MBON, we would then record a net inhibitory response. Given the complexity surrounding these experiments, we decided to focus our attention on activating the cholinergic MBON and recording in LAL and FSB.